# GREED: A Neural Framework for Learning Graph Distance Functions

**Rishabh Ranjan, Siddharth Grover**
Department of Computer Science & Engineering, IIT Delhi, India
{rishabh.ranjan.cs118, siddharth.grover.cs118}@cse.iitd.ac.in

**Sourav Medya**
Department of Computer Science
University of Illinois, Chicago, USA
medya@uic.edu

**Venkat Chakravarthy, Yogish Sabharwal**
IBM Research
Delhi, India
{vechakra, ysabharwal}@in.ibm.com

**Sayan Ranu**
Department of Computer Science & Engineering and Yardi School of AI (Jointly), IIT Delhi, India
sayanranu@cse.iitd.ac.in

## Abstract

Among various distance functions for graphs, graph and subgraph edit distances (GED and SED respectively) are two of the most popular and expressive measures. Unfortunately, exact computations for both are NP-hard. To overcome this computational bottleneck, neural approaches to learn and predict edit distance in polynomial time have received much interest. While considerable progress has been made, there exist limitations that need to be addressed. First, the efficacy of an approximate distance function lies not only in its approximation accuracy, but also in the preservation of its properties. To elaborate, although GED is a metric, its neural approximations do not provide such a guarantee. This prohibits their usage in higher order tasks that rely on metric distance functions, such as clustering or indexing. Second, several existing frameworks for GED do not extend to SED due to SED being asymmetric. In this work, we design a novel siamese graph neural network called GREED, which through a carefully crafted inductive bias, learns GED and SED in a property-preserving manner. Through extensive experiments across 10 real graph datasets containing up to 7 million edges, we establish that GREED is not only more accurate than the state of the art, but also up to 3 orders of magnitude faster. Even more significantly, due to preserving the triangle inequality, the generated embeddings are indexable and consequently, even in a CPU-only environment, GREED is up to 50 times faster than GPU-powered baselines for graph / subgraph retrieval.

## 1 Introduction and Related Work

A distance function on any dataset, including graphs, is a fundamental operator. Among several distance measures on graphs, *edit distance* is one of the most powerful and popular mechanisms [30, 51, 49, 20]. Edit distance can be posed in two forms: *graph edit distance* (GED) and *subgraph edit distance* (SED). Given two graphs $\mathcal{G}_1$ and $\mathcal{G}_2$, $\text{GED}(\mathcal{G}_1, \mathcal{G}_2)$ returns the minimum cost of *edits* needed to convert $\mathcal{G}_1$ to $\mathcal{G}_2$, i.e., for $\mathcal{G}_1$ to become *isomorphic* to $\mathcal{G}_2$. An edit can be the addition or deletion of edges and nodes, or the replacement of edge or node labels, with an associated cost. In $\text{SED}(\mathcal{G}_1, \mathcal{G}_2)$,

36th Conference on Neural Information Processing Systems (NeurIPS 2022).

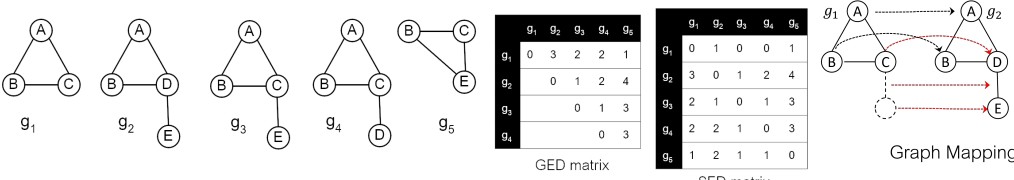

Figure 1: **A sample set of graphs ($g_1$-$g_5$), their corresponding GED and SED matrices and an example of a graph mapping from $g_1$ to $g_2$. The dashed nodes and edges in the mapping represent dummy nodes and edges. The red arrows denote either insertion or change of label.**

the goal is to identify the minimum cost of edits so that $\mathcal{G}_1$ is a subgraph (*subgraph isomorphic*) of $\mathcal{G}_2$. For examples, see Fig. 1.

GED is typically restricted to graph databases containing small graphs to facilitate distance computation with queries of similar sizes. As an example, given a repository of molecules, and a query molecule, we may want to identify the closest molecule in the repository that is similar to the query [36, 20]. SED, on the other hand, is useful when the database has large graphs and the query is a comparatively smaller graph. As examples, subgraph queries are used on knowledge graphs for analogy reasoning [17]. In PPI and chemical compounds, SED is of central importance to identify functional motifs and binding pockets [42, 20, 16, 38, 35, 37]. Unfortunately, both GED and SED are NP-hard to compute [49, 20]. To mitigate this computational bottleneck, several heuristics [10, 13] and index structures [20, 49, 30, 51] have been proposed. Recently, graph neural networks have been shown to be effective in learning and predicting GED [3, 45, 29, 4, 50, 46, 14, 2]. The basic goal in all these algorithms is to learn a neural model from a training set of graph pairs and their distances, such that, at inference time, given an unseen graph pair, we are able to predict its distance accurately. Other works seek to incorporate non-neural graph matching solvers [40], or generic integer linear programming solvers [33, 34], to learn the graph matching task from natural data such as images in an end-to-end trainable manner. Here, the NP-hardness of the problem is relegated to the non-neural component, and the neural part is only concerned with representation learning, unlike in our setting where we want the neural network to handle both. In the domain of subgraphs, NEUROMATCH [39] and ISONET [41] generate embeddings to detect subgraph isomorphism. NSC [44] generates subgraph level embeddings that can count the number of subgraph instances of a query graph on a target graph. While the progress made is impressive, there is scope to do more.

- **Preservation of theoretical properties:** GED is a *metric* distance function. While SED is not metric due to being asymmetric, it satisfies the *triangle inequality*, *non-negativity*, and *subgraph-identity* (SED = 0 for subgraphs). Metrics (and triangle inequality) exhibit significant computational advantages over non-metrics. Specifically, operations such as clustering [19], nearest neighbor search [15, 21, 43, 12], outlier detection [1] and diameter computation [23] admit efficient algorithms precisely when the objects being studied are embedded in a metric space. Existing neural approaches do not preserve these properties, which limits their usability for these higher order tasks.
- **Indexable embeddings:** Given graph pair $\mathcal{G}_1$ and $\mathcal{G}_2$, neural approaches first embed them into a feature space. Next, they compute a distance on these feature vectors, which is an approximation of the distance in the original graph space. The literature on indexing range and $k$-NN queries over feature vectors is rich [18, 25, 12]. Index structures typically allow sub-linear computation costs with respect to the database size. Unfortunately, none of the existing neural approaches generate indexable feature vectors since they perform *pair-dependent* computations. Specifically, the neural computations on $\mathcal{G}_1$ depend on both $\mathcal{G}_1$ and $\mathcal{G}_2$ (and same for $\mathcal{G}_2$). Consequently, the computations can only be done at query-time and thereby negating the possibility of indexing pre-computed feature space embeddings.
- **Modeling SED:** Prior to this work, there have been no neural approaches for SED. Further, existing neural methods to learning GED cannot easily be adapted to learn SED. While GED is symmetric, SED is not. Several neural architectures for GED have the assumption of symmetry at its core and hence modeling SED is non-trivial [3, 4, 29].
- **Exponential Search Space:** Computing SED($\mathcal{G}_1, \mathcal{G}_2$) conceptually requires us to compare the query graph $\mathcal{G}_1$ with the exponentially many subgraphs of the target graph $\mathcal{G}_2$. Therefore, it is imperative that the model has an efficient and effective mechanism to prune the search space without compromising on the prediction accuracy.

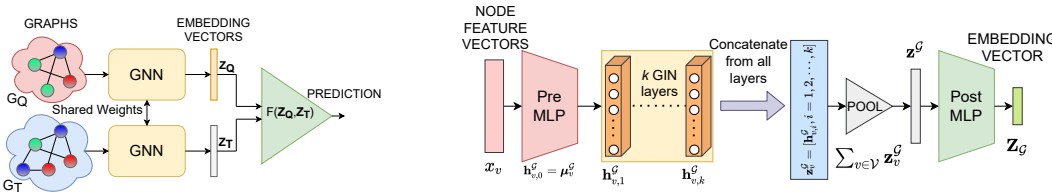

(a) Siamese architecture of GREED        (b) The GNN component in GREED.

Figure 2: **The architecture of GREED.**

In this work, we address the above limitations through the following contributions.

- **Novel neural architecture:** We address the above mentioned challenges through a novel architecture called GREED: GRaph Embeddings for Edit Distances. GREED utilizes a *siamese graph isomorphism network* [47] to embed graphs in a *pair-independent* fashion. A simple, but theoretically well-characterized, function on this embedding space predicts the SED and GED. The carefully crafted prediction function serves as an *inductive bias* for the model, which, in addition to enabling high generalization accuracy, preserves the metric property of GED and the triangle inequality of SED in the embedding space.
- **Indexable embeddings:** Owing to pair-independent embeddings and preservation of the triangle inequality over the embedding space for both SED and GED, GREED can exploit the rich literature on index structures [18, 25, 12] to boost efficiency.
- **Accurate, Fast and Scalable:** Extensive experiments on real graph datasets containing up to a million nodes establish that GREED is more accurate in both GED and SED when compared to the state of the art algorithms and is more than 3 orders of magnitude faster in range and $k$-NN queries. Furthermore, owing to indexable embeddings, even in a CPU-only environment, GREED is up to 50 times faster than the closest baseline run on a GPU.

## 2 Preliminaries and Problem Formulation

We denote a labeled undirected graph as $\mathcal{G} = (\mathcal{V}, \mathcal{E}, \mathcal{L})$ where $\mathcal{V}$ is the node set, $\mathcal{E}$ is the edge set and $\mathcal{L} : \mathcal{V} \cup \mathcal{E} \to \Sigma$ is the labeling function over nodes and edges. $\Sigma$ is the universe of all labels and contains a special empty label $\epsilon$. $\mathcal{L}(v)$ and $\mathcal{L}(e)$ denote the labels of node $v$ and edge $e$ respectively. $\mathcal{G}_1 \subseteq \mathcal{G}_2$ denotes that $\mathcal{G}_1$ is a *subgraph* of $\mathcal{G}_2$.

The problem of learning GED and SED [20] is defined as follows.

**Problem 1 (Learning GED/SED)** *Given a training set of tuples of the form $\langle \mathcal{G}_1, \mathcal{G}_2, \text{GED}(\mathcal{G}_1, \mathcal{G}_2) \rangle$ (or $\mathcal{G}_1, \mathcal{G}_2, \text{SED}(\mathcal{G}_1, \mathcal{G}_2) \rangle$), learn a neural model to predict $\text{GED}(\mathcal{Q}_1, \mathcal{Q}_2)$ (or $\text{SED}(\mathcal{Q}_1, \mathcal{Q}_2)$) on unseen graphs $\mathcal{Q}_1$ and $\mathcal{Q}_2$.*

For details on the exact definition of GED and SED, we refer to the appendix (App. A).

### 2.1 Properties of GED and SED

**Theorem 1** *Let $\widehat{d} : \Sigma \times \Sigma \to \mathbb{R}_0^+$ be a distance function over $\Sigma$, where (i) $\widehat{d}(\ell_1, \ell_2) = 0$ if $\ell_1 = \epsilon$, and (ii) $\widehat{d}(\ell_1, \ell_2) = d(\ell_1, \ell_2)$ otherwise; the following holds: $\text{SED}(\mathcal{G}_1, \mathcal{G}_2) = \widehat{\text{GED}}(\mathcal{G}_1, \mathcal{G}_2)$, where $\widehat{\text{GED}}$ denotes GED with $\widehat{d}$ as the label set distance function. In simple words, the SED between two graphs is equivalent to GED with a label set distance function where we ignore insertion costs.*

PROOF. See App. B.1.

**Observation 1** *GED satisfies the triangle inequality if the distance function $d$ over label set $\Sigma$ satisfies the triangle inequality [20]. As defined in the paragraph following Def. 2, $d$ satisfies the triangle inequality [20]. Furthermore, it is trivial to see that GED is symmetric, non-negative and satisfies identity as long as $d$ satisfies the analogues. Hence, GED with distance function $d$ is metric.*

**Theorem 2** *SED is not metric due to violating the properties of symmetry and identity. However, it satisfies the triangle inequality, i.e., $\text{SED}(\mathcal{G}_1, \mathcal{G}_3) \leq \text{SED}(\mathcal{G}_1, \mathcal{G}_2) + \text{SED}(\mathcal{G}_2, \mathcal{G}_3)$.*

PROOF: See App. B.2 for details.

**Observation 2** *Computing* GED *and* SED *is NP-hard [49].*

Hereon, we use GED as the illustrative distance function being modeled. The architecture trivially extends to SED. The specific places that need separate treatment will be discussed explicitly.

## 3 GREED: The Proposed Architecture

Fig. 2 presents the architecture of GREED. The input to our learning framework is a pair of graphs $\mathcal{G}_{\mathcal{Q}}$ (query), $\mathcal{G}_{\mathcal{T}}$ (target) along with the supervision data $\text{GED}(\mathcal{G}_Q, \mathcal{G}_T)$ (or $\text{SED}(\mathcal{G}_Q, \mathcal{G}_T)$). Our objective is to train a model that can predict GED on unseen query and target graphs. The design of our model must be cognizant of the fact that computing GED is NP-hard and high quality training data is scarce. Thus, we use a *Siamese* architecture [11], where there are *two* networks with *shared* parameters applied to two inputs independently to compute representations.

### 3.1 Siamese Graph Neural Network

As depicted in Fig. 2a, we use a siamese graph neural network (GNN) with shared parameters to embed both $\mathcal{G}_{\mathcal{Q}}$ and $\mathcal{G}_{\mathcal{T}}$. While one could use two different GNN models for the query and the target, this design increases the model parameters and consequently, the training time. Furthermore, an architecture with higher number of parameters also requires larger amount of training data, which is difficult due to GED being NP-hard.

Fig. 2b focuses on the GNN component of GREED. We next discuss each of its individual components.

**Pre-MLP:** $\mathbf{x}_v$ in Fig. 2b is a one-hot encoding of the categorical node labels. The dimension of the one-hot vector increases linearly with the number of labels in a graph database and hence can be very large. The primary-job of the pre-mlp is to reduce it to a desirable dimension size through the operation $\boldsymbol{\mu}_v^{\mathcal{G}} = \text{MLP}(\mathbf{x}_v)$. Indeed, a similar effect may also be obtained by directly feeding the one-hot encoding to the first layer of GIN. However, since a GIN constructs embeddings by incorporating both structure and label information, one may desire a different dimensionality in the GIN layers. Hence, the pre-mlp is motivated more from a conceptual separation of its task to that of GIN rather than a purely performance point of view (See App. G). We do not explicitly model edge labels in our experiments. GREED can easily be extended to edge labels by using GINE [22] instead of GIN.

**Graph Isomorphism Network (GIN):** GIN [47] consumes the information from the Pre-MLP to learn hidden representations that encode both the graph structure as well as the node feature information. GIN is as powerful as the *Weisfeiler-Lehman (WL) graph isomorphism test* [26] in distinguishing graph structures. Since our goal is to accurately characterize graph topology and learn similarity, GIN emerges as the natural choice. GIN develops its expressive power by using an *injective* aggregation function. Specifically, in the initial layer, each node $v$ in graph $\mathcal{G}$ is characterized by the representation learned by the MLP, i.e., $\mathbf{h}_{v,0}^{\mathcal{G}} = \boldsymbol{\mu}_v^{\mathcal{G}}$. Subsequently, in each hidden layer $i$, we learn an embedding through the following transformation.

$$\mathbf{h}_{v,i}^{\mathcal{G}} = \text{MLP}\left((1 + \epsilon^i) \cdot \mathbf{h}_{v,i-1}^{\mathcal{G}} + \sum_{u \in \mathcal{N}_{\mathcal{G}}(v)} \mathbf{h}_{u,i-1}^{\mathcal{G}}\right) \tag{1}$$

Here, $\epsilon^i$ is a layer-specific learnable parameter, $\mathcal{N}_{\mathcal{G}}(v)$ is one-hop neighbourhood of the node $v$, and $\mathbf{h}_{v,0}^{\mathcal{G}} = \boldsymbol{\mu}_v^{\mathcal{G}}$. The $k$-th layer embedding is $\mathbf{h}_{v,k}^{\mathcal{G}}$, where $k$ is final hidden layer.

**Concatenation, Pool and Post-MLP:** Intuitively, $\mathbf{h}_{v,i}^{\mathcal{G}}$ captures a feature-space representation of the $i$-hop neighborhood of $v$. Typically, GNNs operate on node or edge level predictive tasks, such as node classification or link prediction, and hence, the node representations are passed through an MLP for the final prediction task. In our problem, we need to capture a graph level representation. Furthermore, the representation should be rich enough to also capture the various subgraphs within the input graph so that SED can be predicted accurately. To fulfil these requirements, we first *concatenate* the representation of a node across *all* hidden layers, i.e., the final node embedding is $\mathbf{z}_v^{\mathcal{G}} = \text{CONCAT}\left(\mathbf{h}_{v,i}^{\mathcal{G}}, \forall i \in \{1, 2, \cdots, k\}\right)$. This allows us to capture a multi-granular view of the subgraphs centered on $v$ at different radii in the range $[1, k]$. Next, to construct the graph-level representation, we perform a sum-pool, which adds the node representations to give a single vector. This information is then fed to the Post-MLP to enable post-processing. Mathematically:

$$\mathbf{Z}_{\mathcal{G}} = \text{MLP}(\mathbf{z}^{\mathcal{G}}) = \text{MLP}\left(\sum_{v \in \mathcal{V}} \mathbf{z}_v^{\mathcal{G}}\right) \tag{2}$$

**GED and SED Prediction:** The final task is to predict the GED (and SED) as a function of query graph embedding $\mathbf{Z}_{\mathcal{G}_{\mathcal{Q}}}$ and target graph embedding $\mathbf{Z}_{\mathcal{G}_{\mathcal{T}}}$. The natural choice would be to feed these embeddings into another MLP to learn $\text{GED}(\mathbf{Z}_{\mathcal{G}_{\mathcal{Q}}}, \mathbf{Z}_{\mathcal{G}_{\mathcal{T}}})$. This MLP can then be trained jointly with the graph embedding model in an *end-to-end* fashion. However, an MLP prediction does not have any theoretical guarantees with respect to the preservation of metric properties of GED and the triangle inequality of SED. We, therefore, focus on learning prediction functions $\mathcal{F}_g\left(\mathbf{Z}_{\mathcal{G}_{\mathcal{Q}}}, \mathbf{Z}_{\mathcal{G}_{\mathcal{T}}}\right)$ and $\mathcal{F}_s\left(\mathbf{Z}_{\mathcal{G}_{\mathcal{Q}}}, \mathbf{Z}_{\mathcal{G}_{\mathcal{T}}}\right)$ for GED and SED respectively, such that they are accurate and respects the desirable properties from the original graph space. As we will empirically substantiate in § 4.5, the inductive bias injected through the prediction functions also lead to more effective learning over low volumes of training data than an MLP.

### 3.1.1 GED

We require the following four properties to ensure that the prediction is also a metric.

$$\mathcal{F}_g\left(\mathbf{Z}_{\mathcal{G}_{\mathcal{Q}}}, \mathbf{Z}_{\mathcal{G}_{\mathcal{T}}}\right) \geq 0 \tag{3}$$

$$\mathcal{F}_g\left(\mathbf{Z}_{\mathcal{G}_{\mathcal{Q}}}, \mathbf{Z}_{\mathcal{G}_{\mathcal{T}}}\right) = 0 \iff \forall i : \mathbf{Z}_{\mathcal{G}_{\mathcal{Q}}}[i] = \mathbf{Z}_{\mathcal{G}_{\mathcal{T}}}[i] \tag{4}$$

$$\mathcal{F}_g\left(\mathbf{Z}_{\mathcal{G}_{\mathcal{Q}}}, \mathbf{Z}_{\mathcal{G}_{\mathcal{T}}}\right) = \mathcal{F}_g\left(\mathbf{Z}_{\mathcal{G}_{\mathcal{T}}}, \mathbf{Z}_{\mathcal{G}_{\mathcal{Q}}}\right) \tag{5}$$

$$\mathcal{F}_g\left(\mathbf{Z}_{\mathcal{G}_{\mathcal{Q}}}, \mathbf{Z}_{\mathcal{G}_{\mathcal{T}}}\right) \leq \mathcal{F}_g\left(\mathbf{Z}_{\mathcal{G}_{\mathcal{Q}}}, \mathbf{Z}_{\mathcal{G}'}\right) + \mathcal{F}_g\left(\mathbf{Z}_{\mathcal{G}'}, \mathbf{Z}_{\mathcal{G}_{\mathcal{T}}}\right) \tag{6}$$

To achieve this, we establish an important connection of metrics on vector spaces to norms. Every norm $\|.\|$ gives a metric $(\mathbf{x}, \mathbf{y}) \mapsto \|\mathbf{x} - \mathbf{y}\|$. Moreover for a metric, there exists a norm $\|.\|$ such that the metric can be expressed as $(\mathbf{x}, \mathbf{y}) \mapsto \|\mathbf{x} - \mathbf{y}\|$, *iff* the metric is *translation invariant* and *homogeneous*. Thus, we add these properties to the desiderata for $\mathcal{F}_g$:

$$\mathcal{F}_g\left(\mathbf{Z}_{\mathcal{G}_{\mathcal{Q}}} + \mathbf{k}, \mathbf{Z}_{\mathcal{G}_{\mathcal{T}}} + \mathbf{k}\right) = \mathcal{F}_g\left(\mathbf{Z}_{\mathcal{G}_{\mathcal{Q}}}, \mathbf{Z}_{\mathcal{G}_{\mathcal{T}}}\right), \forall \mathbf{k} \in \mathbb{R}^d \tag{7}$$

$$\mathcal{F}_g\left(r\mathbf{Z}_{\mathcal{G}_{\mathcal{Q}}}, r\mathbf{Z}_{\mathcal{G}_{\mathcal{T}}}\right) = |r|\mathcal{F}_g\left(\mathbf{Z}_{\mathcal{G}_{\mathcal{Q}}}, \mathbf{Z}_{\mathcal{G}_{\mathcal{T}}}\right), \forall r \in \mathbb{R} \tag{8}$$

Armed with these observations, we define the class of functions that may be used for $\mathcal{F}_g$.

**Observation 3** *$\mathcal{F}_g$ may be defined as any function $(x, y) \mapsto \|x - y\|$ for some norm $\|.\|$ over the vector space $\mathbb{R}^d$ such that $\mathcal{F}_g$ satisfies Eqs. 7 - 8.*

The $L_p$ norm satisfies Obs. 3. Hence, we define $\mathcal{F}_g$ as:

$$\mathcal{F}_g\left(\mathbf{Z}_{\mathcal{G}_{\mathcal{Q}}}, \mathbf{Z}_{\mathcal{G}_{\mathcal{T}}}\right) = \|\mathbf{Z}_{\mathcal{G}_{\mathcal{Q}}} - \mathbf{Z}_{\mathcal{G}_{\mathcal{T}}}\|_p \tag{9}$$

In our implementation we use the $L_2$ norm. Finally, the parameters of the entire model are learned by minimizing the mean squared error (here $\mathbb{T}$ is the training set).

$$\mathscr{L} = \frac{1}{|\mathbb{T}|} \sum_{\forall \langle \mathcal{G}_{\mathcal{Q}}, \mathcal{G}_{\mathcal{T}} \rangle \in \mathbb{T}} \left(\mathcal{F}_g\left(\mathbf{Z}_{\mathcal{G}_{\mathcal{Q}}}, \mathbf{Z}_{\mathcal{G}_{\mathcal{T}}}\right) - \text{GED}\left(\mathcal{G}_{\mathcal{Q}}, \mathcal{G}_{\mathcal{T}}\right)\right)^2 \tag{10}$$

**Intuition:** Regardless of the graph representations generated by our model, $\mathcal{F}_g$ ensures that the predicted distance is a metric. One the other hand, by training the model to produce embeddings $\mathbf{Z}_{\mathcal{G}_{\mathcal{Q}}}$ and $\mathbf{Z}_{\mathcal{G}_{\mathcal{T}}}$ such that $\mathcal{F}_g\left(\mathbf{Z}_{\mathcal{G}_{\mathcal{Q}}}, \mathbf{Z}_{\mathcal{G}_{\mathcal{T}}}\right) \approx \text{GED}\left(\mathbf{Z}_{\mathcal{G}_{\mathcal{Q}}}, \mathbf{Z}_{\mathcal{G}_{\mathcal{T}}}\right)$, we enforce a rich structure on the embedding space such that $\mathcal{F}_g$ is also accurate. Thus, $\mathcal{F}_g$ injects an inductive bias satisfying the dual needs of accuracy and preservation of original space properties.

### 3.1.2 SED

SED satisfies non-negativity and triangle inequality. Following a similar reasoning as above, we define $\mathcal{F}_s$ as follows:

$$\mathcal{F}_s\left(\mathbf{Z}_{\mathcal{G}_{\mathcal{Q}}}, \mathbf{Z}_{\mathcal{G}_{\mathcal{T}}}\right) = \|ReLU(\mathbf{Z}_{\mathcal{G}_{\mathcal{Q}}} - \mathbf{Z}_{\mathcal{G}_{\mathcal{T}}})\|_2 = \left\|\max\left\{0, \mathbf{Z}_{\mathcal{G}_{\mathcal{Q}}} - \mathbf{Z}_{\mathcal{G}_{\mathcal{T}}}\right\}\right\|_2 \tag{11}$$

Intuitively, for those co-ordinates where the value of $\mathbf{Z}_{\mathcal{G}_{\mathcal{Q}}}$ is greater than $\mathbf{Z}_{\mathcal{G}_{\mathcal{T}}}$, a distance penalty is accounted by $\mathcal{F}_s$ in terms of how much those values differ; otherwise $\mathcal{F}_s$ considers 0. This follows the intuition that the SED accounts for those features of $\mathcal{G}_{\mathcal{Q}}$ that are not in $\mathcal{G}_{\mathcal{T}}$. Moreover, consistent with SED, the additional features in $\mathcal{G}_{\mathcal{T}}$ that are not in $\mathcal{G}_{\mathcal{Q}}$, do not incur any cost.

**Lemma 1** *The following properties hold on predicted* SED.

1. $\mathcal{F}_s\left(\mathbf{Z}_{\mathcal{G}_{\mathcal{Q}}}, \mathbf{Z}_{\mathcal{G}_{\mathcal{T}}}\right) \geq 0$
2. $\mathcal{F}_s\left(\mathbf{Z}_{\mathcal{G}_{\mathcal{Q}}}, \mathbf{Z}_{\mathcal{G}_{\mathcal{T}}}\right) = 0 \iff \mathbf{Z}_{\mathcal{G}_{\mathcal{Q}}} \leq \mathbf{Z}_{\mathcal{G}_{\mathcal{T}}}$
3. $\mathcal{F}_s\left(\mathbf{Z}_{\mathcal{G}_{\mathcal{Q}}}, \mathbf{Z}_{\mathcal{G}_{\mathcal{T}}}\right) \leq \mathcal{F}_s(\mathbf{Z}_{\mathcal{G}_{\mathcal{Q}}}, \mathbf{Z}_{\mathcal{G}_{\mathcal{T}'}}) + \mathcal{F}_s(\mathbf{Z}_{\mathcal{G}_{\mathcal{T}'}}, \mathbf{Z}_{\mathcal{G}_{\mathcal{T}}})$

PROOF. Properties (1) and (2) follow from the definition of $\mathcal{F}$ itself. Property (3) follows from the fact that we take the $L_2$ norm. Formally, we state it as follows.

**Lemma 2** $\mathcal{F}_s\left(\mathbf{Z}_{\mathcal{G}_{\mathcal{Q}}}, \mathbf{Z}_{\mathcal{G}_{\mathcal{T}}}\right) \leq \mathcal{F}_s(\mathbf{Z}_{\mathcal{G}_{\mathcal{Q}}}, \mathbf{Z}_{\mathcal{G}_{\mathcal{T}'}}) + \mathcal{F}_s(\mathbf{Z}_{\mathcal{G}_{\mathcal{T}'}}, \mathbf{Z}_{\mathcal{G}_{\mathcal{T}}})$, where $\mathcal{F}_s(\mathbf{x}, \mathbf{y}) = \|ReLU(\mathbf{x} - \mathbf{y})\|$ *for any monotonic norm* $\|.\|$.

PROOF. See App. B.4.

## 3.2 Characterization of GREED

**Importance of pair-independence and siamese architecture:** A pair-independent siamese architecture enables theoretical guarantees for GREED by constraining the model to learn a single mapping from graph space to embedding space for both query and target. Despite these restrictions, GREED outperforms prior works which freely use cross-graph information and don't provide theoretical guarantees. This further confirms that the siamese architecture is a useful prior.

**Complexity Analysis:** The complexity of GED and SED inference in GREED is *linear* in the number of nodes and edges in the query and target graphs (See App. C for derivation). This computation cost is drastically lower than the factorial computation cost of optimal GED and SED. With respect to neural methods for graph similarity [3, 4, 29, 45, 50], all have at least quadratic computation cost, i.e., $O(|\mathcal{V}|^2)$.

**Indexing Embeddings:** Since the generated embeddings for both GED and SED satisfy triangle inequality, they are indexable leading to fast querying times. We develop an index structure to exploit this property. Due to space limitations, the details are included in App. D. In addition, we also design a *neighborhood decomposition* scheme, which enables fast pruning of the exponential search space § D.4. In § 4.3, we empirically analyze the impact of index structures on querying time.

# 4 Empirical Evaluation

In this section, we establish the following:

- **Efficacy:** GREED is more accurate than the state of the art approaches for both GED and SED.
- **Efficiency:** GREED is orders of magnitude faster than existing approaches and scales well to graphs with millions of nodes.
- **Scalability:** Pair-independence and indexability further enhances the scalability of GREED and enables it to be run on CPU-only platforms.

Our code base and datasets are available at https://github.com/idea-iitd/greed.

## 4.1 Experimental Setup

We use a machine with an Intel Xeon Gold 6142 processor and GeForce GTX 1080 Ti GPU for all our experiments.

**Datasets:** Table A lists the datasets used for benchmarking. Further details on the dataset semantics are provided in the App. E. We include a mixture of both *graph databases* (#graphs >1), as well as *single large* graphs (#graphs = 1). `Linux` and `IMDB` are unlabeled. We note that this is the first study to evaluate neural graph distance approaches on million-scale graphs.

**Baselines:** To evaluate performance in **GED**, we compare with SIMGNN [3], GENN-A* [45], H²MN[50] and GOTSIM [14]. These are the most recent neural frameworks and have shown better efficacy than other neural approaches such as SIMGNN [3], GRAPHSIM [4], and GMN [29].

For **SED**, no neural approaches exist. However, H²MN and SIMGNN can be trained by replacing GED with SED along with minor modifications in training. NSC [31] is a method for counting subgraphs using graph embeddings. Since this is a related operation, we use NSC as a baseline by changing the loss function to minimize the RMSE between true and predicted SED. We also use NEUROMATCH [39] as a baseline, which was originally designed to detect subgraph isomorphism. While NEUROMATCH cannot predict SED, it generates a *violation score*, which can be interpreted as

| Methods | AIDS' | Linux | IMDB |
|---|---|---|---|
| **GREED** | **0.796** | 0.415 | **6.734** |
| H$^2$MN | 0.994 | 0.734 | 86.077 |
| GENN-A* | 0.907 | **0.267** | NA |
| GOTSIM | 0.996 | 0.574 | 37.831 |
| SIMGNN | 1.037 | 0.666 | 66.250 |
| Branch | 3.322 | 2.474 | 6.875 |
| MIP-F2 | 2.929 | 1.245 | 82.124 |

(a) Prediction of GED

| Methods | Dblp | Amazon | PubMed | CiteSeer | Cora_ML | Protein | AIDS |
|---|---|---|---|---|---|---|---|
| **GREED** | **0.964** | **0.495** | **0.728** | **0.519** | **0.635** | **0.524** | **0.512** |
| H$^2$MN | 1.470 | 1.294 | 1.213 | 1.502 | 1.446 | 0.941 | 0.749 |
| NSC | NA | 2.141 | 1.095 | 1.66 | 1.661 | 0.662 | 0.562 |
| SIMGNN | 1.482 | 2.810 | 1.322 | 1.781 | 1.289 | 1.223 | 0.696 |
| Branch | 2.917 | 4.513 | 2.613 | 3.161 | 3.102 | 2.391 | 1.379 |
| MIP-F2 | 3.427 | 5.595 | 3.399 | 4.474 | 3.871 | 2.249 | 1.537 |

(b) Prediction of SED.

Table 1: **(a) Datasets. (b-c) RMSE scores (lower is better) in (a) GED and (b) SED. GENN-A*** **does not scale on graphs beyond** 10 **nodes and hence results in** IMDB **are not reported. NSC does not scale in** Dblp **due to memory consumption.**

the likelihood of the query being subgraph isomorphic to the target. The violation score can be used as a proxy for SED and used in ranking of $k$-NN ($k$-NearestNeighbour) queries. Thus, NEUROMATCH comparisons are limited to $k$-NN queries on SED. GMN [29], GRAPHSIM [4], GOTSIM [14] and GENN-A* are not included since they cannot be easily adapted for SED. See App. F for details.

In the **non-neural** category, we use *mixed integer programming* based method MIP-F2 [27] with a time bound of 0.1 seconds per pair for both GED and SED. MIP-F2 provides the optimal solution given infinite time. We also compare with BRANCH [5], which achieves an excellent trade-off between accuracy and time [6]. BRANCH uses *linear sum assignment problem with error-correction* (LSAPE) to process the search space. We use GEDLIB's [7] implementation of these methods.
**Training (and Test) Data Generation:** For GED, we use $\langle query, target \rangle$ graph pairs from IMDB, AIDS', and Linux. Our setup is identical to SIMGNN [3] and H$^2$MN [50].

For SED, the target graphs are taken from datasets listed in Table A. For the query graph, in AIDS, we use known *functional groups* [38]. In the rest of the graph datasets, queries are sampled by performing a random BFS traversal (depth up to 5). Table A shows the average query sizes ($|\mathcal{V}_\mathcal{Q}|$, $|\mathcal{E}_\mathcal{Q}|$).We use *mixed integer programming* method F2 [27] implemented in GEDLIB [7] with a large time limit to generate ground-truth data.
**Train-Validation-Test:** We use $100K$ query-target pairs for training and $10K$ pairs each for validation and test. All models are trained till validation loss is minimized or there is less than $0.05\%$ change in validation loss over a number of extended epochs. For GREED, we set the number of layers in GIN to 8. The hidden layer dimension is set to 64. For all baselines, we use the default parameters suggested by the authors.

## 4.2 Prediction Accuracy of SED and GED

Tables 1a and 1b present the accuracy of all techniques on GED and SED in terms of *Root Mean Square Error (RMSE)*. GREED outperforms all other techniques in 9 out of 10 settings across GED and SED. While H$^2$MN and NSC are the second best performers in SED, GENN-A* performs well in GED. GENN-A*, however, is extremely slow and does not scale on graphs of size beyond 10. H$^2$MN, thus, provides the second best balance between efficacy and efficiency after GREED. The gap in accuracy is the highest in IMDB for GED, where GREED is more than 10 times better than the neural baselines. IMDB graphs are significantly denser and larger than AIDS' or Linux. Thus, computing the optimal GED is harder. While all techniques have higher errors in IMDB, the deterioration is more severe in the baselines indicating that GREED scales better with graph sizes.
 **Impact of Query Size:** We next investigate how the accuracy varies against the query size. Intuitively, the task gets harder with query size since the combinatorial space of possible maps increases exponentially. For this analysis, we compare GREED with H$^2$MN in IMDB and Dblp for GED and SED respectively. GENN-A* fails to scale on both datasets.

In Fig. 3, we plot the *heat map* of RMSE against query graph size. In this plot, each dot corresponds to a query graph $\mathcal{G}_\mathcal{Q}$. The co-ordinate of a query is $(\text{GED}(\mathcal{G}_\mathcal{Q}, \mathcal{G}_\mathcal{T}), |\mathcal{V}_\mathcal{Q}|)$ (analogously defined for SED). The color of a dot represents the RMSE; the darker the color, the higher is the RMSE. When we compare the heat maps of GREED with H$^2$MN, we observe that H$^2$MN is noticeably darker. Furthermore, the concentration of dark colors is noticeably higher on the upper-right corner indicating deterioration with larger query sizes and higher distance values. This indicates that GREED scales better with query sizes and distances.

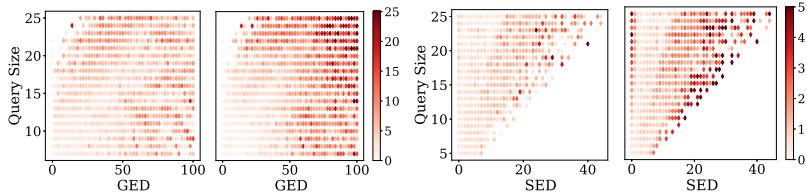

**(a)** GREED, IMDB **(b)** H$^2$MN, IMDB **(c)** GREED, Dblp **(d)** H$^2$MN, Dblp

Figure 3: **Heat Map of RMSE in (a-b) GED and (c-d) SED against query size in IMDB and Dblp.**

| Methods | AIDS' | Linux | IMDB |
|---|---|---|---|
| GREED | **0.80** | 0.89 | **0.87** |
| H$^2$MN | 0.74 | 0.88 | 0.80 |
| GENN-A$^*$ | 0.75 | **0.90** | NA |
| SIMGNN | 0.72 | 0.86 | 0.67 |

(a) Ranking in GED.

| Methods | PubMed | CiteSeer | Cora_ML | Protein | AIDS |
|---|---|---|---|---|---|
| GREED | **0.90** | **0.90** | **0.91** | **0.75** | **0.80** |
| H$^2$MN | 0.87 | 0.88 | 0.88 | 0.70 | 0.72 |
| NSC | 0.89 | 0.88 | 0.88 | 0.74 | 0.78 |
| SIMGNN | 0.85 | 0.87 | 0.86 | 0.63 | 0.73 |
| NEUROMATCH | 0.70 | 0.75 | 0.73 | 0.57 | 0.59 |

(b) Ranking in SED.

Table 2: **Kendall's tau scores (higher is better).**

**Visualization:** A case study to visually illustrate the efficacy of GREED is provided in App. I.

**Range and k-NN queries:** To quantify performance in range and $k$-NN queries, we measure *F1-score (Range query)* and *Kendalls's tau (k-NN)* [24] of the predicted answer set, when compared against the ground truth. In Figs. 4a-4h, we measure the performance in range queries. In SED, GREED consistently outperforms all baselines in F1-score. In GED, the trend remains similar. Although, GENN-A$^*$ outperforms GREED for a brief region in Linux, overall, GREED has the highest F1-score. We also note the GENN-A$^*$ is not included in Fig. 4h since it fails to scale on IMDB. In $k$-NN queries (Tables 2a and 2b), GREED outperforms all algorithms in SED. In GED, similar to the trend in range queries, GREED is the dominant method and GENN-A$^*$ marginally outperforms GREED in Linux.

### 4.3 Efficiency

Tables 3a-3b present the inference times per $10K$ query-target pairs. In this experiment, we do not index embeddings by GREED so that the comparison unearths the raw difference in computation efficiency of solely the neural architectures. As visible, GREED is up to 1800 times faster than the non-neural baselines and up to 10 to 20 times faster than H$^2$MN, the current state of the art in GED prediction. Also note that GENN-A$^*$ is exorbitantly slow (Table 3a). GENN-A$^*$ is slower since it not only predicts the GED but also the alignment via an A$^*$ search. While the alignment information is indeed useful, computing this information across all graphs in the database may generate redundant information since an user is typically interested only on a small minority of graphs that are in the answer set. In App. H, we discuss this issue in detail.

### 4.4 Pair-independence and Indexability

Here, we showcase how pair-independent embeddings, and ensuring triangle inequality leads to further boost in scalability. For this experiment, we use the three largest datasets of PubMed, Amazon and Dblp. For each dataset, we pre-compute GREED embeddings of all database graphs by exploiting pair-independent embeddings. Such pre-computation is not possible in the neural or non-neural baselines. Furthermore, since the predictions of GREED satisfy triangle inequality, we index the pre-computed embeddings of the database graphs as discussed in § D.1. Consequently, for GREED, we only need to embed the query graph and evaluate $\mathcal{F}$ to make predictions at query time. Table 3c presents the results on range and 10-NN queries. When computations are done on a GPU, GREED is more than 1000 times faster than H$^2$MN. In the absence of a GPU, H$^2$MN is practically infeasible since expensive pair-dependent computations are done at query time. In contrast, even on a CPU, through indexing, GREED is $\approx 50$ times faster than GPU-based H$^2$MN. Note that indexing enables up to 3-times speed-up on GREED over linear scan, which demonstrates the gain from ensuring triangle inequality. These results establish that GREED breaks new ground in scalability of neural

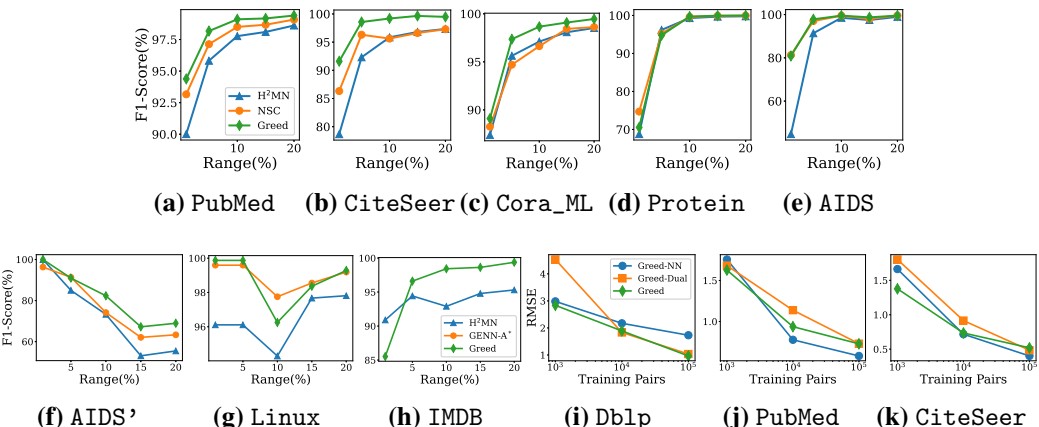

**(a)** PubMed  **(b)** CiteSeer **(c)** Cora_ML  **(d)** Protein  **(e)** AIDS

**(f)** AIDS'  **(g)** Linux  **(h)** IMDB  **(i)** Dblp  **(j)** PubMed  **(k)** CiteSeer

Figure 4: **F1-score in range queries on SED (a-e) and GED (f-h). The range threshold is set as a percentage of the max distance observed in the test set. The legend for Figs. (a)-(e) is provided in (a) and for (f-h) is provided in (h). (i-k) Ablation study to analyze the impact of siamese architecture and function $\mathcal{F}$. The legend for Figs. (i)-(k) is provided in (i).**

| Methods | AIDS' | Linux | IMDB | | Methods | Dblp | Amazon | PubMed | CiteSeer | Cora_ML | Protein | AIDS |
|---|---|---|---|---|---|---|---|---|---|---|---|---|
| **GREED** | **0.49** | **0.70** | **0.63** | | **GREED** | **6.84** | **1.46** | **1.30** | **1.28** | **1.25** | **0.86** | **0.84** |
| **H$^2$MN** | 9.50 | 8.74 | 8.83 | | **H$^2$MN** | 44.68 | 23.2 | 25.79 | 27.54 | 29.04 | 19.33 | 9.63 |
| **GENN-A$^*$** | 12190 | 1340 | NA | | **NSC** | NA | 21 | 35.05 | 24.46 | 70.59 | 21 | 4 |
| **BRANCH** | 10.70 | 8.24 | 127.90 | | **SIMGNN** | 109.56 | 47.68 | 39.80 | 39.40 | 40.73 | 39.02 | 43.83 |
| **MIP-F2** | 593.34 | 191.88 | 1173.548 | | **BRANCH** | 626.489 | 79.25 | 99.11 | 155.09 | 132.98 | 52.26 | 12.93 |
| | | | | | **MIP-F2** | 1979.185 | 861.95 | 606.01 | 827.65 | 790.01 | 881.77 | 360.12 |

(a) GED  (b) SED

| Datasets | Range ($\theta = 2$) | | | | 10-NN | | | |
|---|---|---|---|---|---|---|---|---|
| | CPU | | GPU | | CPU | | GPU | |
| | L-Scan | Indexed | L-Scan | H2MN | L-Scan | Indexed | L-Scan | H2MN |
| PubMed | 0.693 | 0.56 | 0.004 | 26.6 | 1.01 | 0.49 | 0.004 | 27.5 |
| Amazon | 9.09 | 5.07 | 0.025 | 371 | 11.3 | 4.75 | 0.027 | 372 |
| Dblp | 48 | 20.9 | 0.070 | 696 | 50.4 | 18.6 | 0.126 | 698 |

(c) Scalability

Table 3: **(a-b) Running times of all methods in seconds per 10k query-target pair. (c) Querying time (s) for SED in the three largest datasets. L-Scan indicates time taken by linear scan in GREED (times differ based on whether executed on CPU or GPU).**

graph distance computations; not only is it faster, it overcomes the barrier of GPU-dependence and hence better suited for low-resource environments.

## 4.5 Ablation Study

In this study, we explore the impact of our inductive biases in learning from low-volume data. We create two variants of GREED: **(1)** GREED-Dual trains the two parallel GNN models separately without weight-sharing, and **(2)** GREED-NN uses an MLP instead of $\mathcal{F}$. Both have strictly better representational capacity than GREED, so are expected to match the performance with infinite data. Figs. 4i-4k present the results on SED. The results of the same experiment on GED is provided in Figs. F in the appendix. The RMSE of GREED is generally better than GREED-Dual, with the difference being more significant at low volumes. This indicates that siamese structure helps. Compared to GREED, GREED-NN achieves marginally better performance at larger train sizes in PubMed and CiteSeer. However, in Dblp, GREED is consistently better. The number of subgraphs in a dataset grows exponentially with the node set size. Hence, an MLP needs growing training data to accurately model the intricacies of this search space. In Dblp, even 100k pairs is not enough to improve upon $\mathcal{F}$. Furthermore, since computing GED and SED is NP-hard, generating large volumes of training data is not desirable. Overall, these trends indicate that $\mathcal{F}$ enables better generalization and scalability with

respect to accuracy. Furthermore, given that its performance is close to an MLP even on high-volume training data, and it enables indexing, the benefits outweigh the marginal reduction in accuracy.

We also observe that generally, GREED-NN performs better than GREED-dual. GREED-NN retains the inductive bias imparted by the Siamese architecture, but ablates the inductive bias of the custom prediction function. The opposite happens in the case of GREED-Dual. The observed phenomenon of GREED-NN generally out-performing GREED-Dual can be interpreted as evidence for the Siamese architecture providing a stronger inductive bias than the custom prediction function.

More ablations studies justifying our choice of GIN and the sum-pool layer are provided in App. G.

### 4.6 Generalization to Unseen Query Distributions in SED

We train the model by sampling queries from the graph database through BFS enumerations. *How does* GREED *generalize to unseen distributions?* Towards that end, we generate queries from the three unseen distributions of **(1)** Random Walks (RW), **(2)** Random Walks with Restarts (RWR), and **(3)** SHADOW [48] (See App. J for details on the sampling strategies). We first note that in AIDS, we use real queries of functional groups, and thus the good performance in AIDS indicates good generalizability. In Table 4a, we more exhaustively analyze this aspect. As visible, the errors remain low. Even more surprisingly, the errors on RW and RWR are better than the train distribution of BFS itself. This indicates good generalization to unseen distributions.

### 4.7 Generalizability to Unseen, Larger Query Sizes:

Generating training data for learning GED and SED is expensive since optimal distance computations are NP-hard. Hence, a desirable property would be to learn from small graphs and then generalize to larger unseen graphs. We evaluate this ability for GREED and H$^2$MN. Table 4b provides the numbers. We notice that although there is some deterioration in the quality for query sizes in the range $[25, 50]$ when compared to the entire set, it is not severe (GREED-50 in Table 4b). However, if the train set only contains queries till size 25 and we deploy the learned model to infer on queries of larger unseen sizes, the drop in quality is significant (GREED-25 in Table 4b). This drop is even more dramatic in H$^2$MN. On the positive side, GREED remains superior to the optimal non-neural approach (MIP-F2) when run with a generous time limit of 60 seconds per query. Overall, this experiment highlights one direction that needs further study and improvement.

## 5 Conclusions, Limitation, and Future Directions

The problem of learning graph distances from their embeddings has seen much interest over the last few years. This thread of research is important since it allows us to overcome the bottleneck of exponential graph alignment space. Our experiments clearly establish GREED as the state of the art for both GED and SED (See App. K for preliminary results on *maximum common subgraph similarity*). In addition, it is significantly faster and provides better theoretical correspondence between properties of the original space and predicted space. One clear direction of future work that emerges from our experiments is that GREED, and existing methods of graph distance learning, do not generalize well to unseen larger query sizes. We hope to address this limitation next.

| Sampler | PubMed | CiteSeer | Amazon |
|---|---|---|---|
| BFS | 0.728 | 0.519 | 0.495 |
| RW | 0.508 | 0.770 | 0.490 |
| RWR | 0.545 | 0.754 | 0.299 |
| SHADOW | 0.966 | 0.753 | 0.830 |

(a) Query distributions

| Method | PubMed | | CiteSeer | | Amazon | |
|---|---|---|---|---|---|---|
| | $\mathcal{V}_Q \in [0, 50]$ | $\mathcal{V}_Q \in [25, 50]$ | $\mathcal{V}_Q \in [0, 50]$ | $\mathcal{V}_Q \in [25, 50]$ | $\mathcal{V}_Q \in [0, 50]$ | $\mathcal{V}_Q \in [25, 50]$ |
| GREED-50 | 1.294 | 1.917 | 0.728 | 0.948 | 0.638 | 0.782 |
| GREED-25 | 2.824 | 4.999 | 4.740 | 9.052 | 1.152 | 1.724 |
| H$^2$MN-50 | 3.133 | 5.112 | 4.9380 | 8.583 | 6.014 | 9.550 |
| H$^2$MN-25 | 7.417 | 13.366 | 10.459 | 19.787 | 5.720 | 9.462 |
| MIP-F2 | 3.507 | 6.278 | 4.831 | 8.505 | 6.454 | 10.293 |

(b) Query size

Table 4: **(a) RMSE on unseen query distributions. BFS (seen) is the baseline to compare against. (b) RMSE against query sizes. GREED-50 indicates GREED trained on a dataset containing queries of size up to** $50$**. GREED-25 is defined analogously.**

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
