# Appendix

## A  GED and SED

The computation of GED relies on a *graph mapping*.

**Definition 1 (Graph Mapping)** *Given two graphs $\mathcal{G}_1$ and $\mathcal{G}_2$, let $\tilde{\mathcal{G}}_1 = (\tilde{\mathcal{V}}_1, \tilde{\mathcal{E}}_1, \tilde{\mathcal{L}}_1)$ and $\tilde{\mathcal{G}}_2 = (\tilde{\mathcal{V}}_2, \tilde{\mathcal{E}}_2, \tilde{\mathcal{L}}_2)$ be obtained by adding dummy nodes and edges (labeled with $\epsilon$) to $\mathcal{G}_1$ and $\mathcal{G}_2$ respectively, such that $|\mathcal{V}_1| = |\mathcal{V}_2|$ and $|\mathcal{E}_1| = |\mathcal{E}_2|$. A node mapping between $\mathcal{G}_1$ and $\mathcal{G}_2$ is a bijection $\pi : \tilde{\mathcal{G}}_1 \to \tilde{\mathcal{G}}_2$ where (i) $\forall v \in \tilde{\mathcal{V}}_1, \pi(v) \in \tilde{\mathcal{V}}_2$ and at least one of $v$ and $\pi(v)$ is not a dummy; (ii) $\forall e = (v_1, v_2) \in \tilde{\mathcal{E}}_1, \pi(e) = (\pi(v_1), (\pi(v_2))) \in \tilde{\mathcal{E}}_2$ and at least one of $e$ and $\pi(e)$ is not a dummy.*

**Example 1** *Fig. 1 shows a graph mapping. Edge mappings can be trivially inferred.*

**Definition 2 (Graph Edit Distance (GED) under mapping $\pi$)** GED *between $\mathcal{G}_1$ and $\mathcal{G}_2$ under $\pi$ is*

$$\text{GED}_\pi(\mathcal{G}_1, \mathcal{G}_2) = \sum_{v \in \tilde{V}_1} d(\mathcal{L}(v), \mathcal{L}(\pi(v))) + \sum_{e \in \tilde{E}_1} d(\mathcal{L}(e), \mathcal{L}(\pi(e))) \tag{12}$$

*where $d : \Sigma \times \Sigma \to \mathbb{R}_0^+$ is a distance function over the label set. $d(\ell_1, \ell_2)$ models an insertion if $\ell_1 = \epsilon$, deletion if $\ell_2 = \epsilon$ and replacement if $\ell_1 \neq \ell_2$ and neither $\ell_1$ nor $\ell_2$ is a dummy.*

We assume $d$ to be a binary function, where $d(\ell_1, \ell_2) = 1$ if $\ell_1 \neq \ell_2$, otherwise, 0.

**Definition 3 (Graph Edit Distance (GED))** GED *is the minimum distance under all mappings.*

$$\text{GED}(\mathcal{G}_1, \mathcal{G}_2) = \min_{\forall \pi \in \Phi(\mathcal{G}_1, \mathcal{G}_2)} \text{GED}_\pi(\mathcal{G}_1, \mathcal{G}_2) \tag{13}$$

$\Phi(\mathcal{G}_1, \mathcal{G}_2)$ *denotes the set of all possible node maps from $\mathcal{G}_1$ to $\mathcal{G}_2$.*

**Definition 4 (Subgraph Edit Distance (SED))** SED *is the minimum* GED *over all subgraphs of $\mathcal{G}_2$.*

$$\text{SED}(\mathcal{G}_1, \mathcal{G}_2) = \min_{\mathcal{S} \subseteq \mathcal{G}_2} \text{GED}(\mathcal{G}_1, \mathcal{S}) \tag{14}$$

**Observation 4** *(i) $\text{GED}(\mathcal{G}_1, \mathcal{G}_2) \geq 0$, (ii) $\text{SED}(\mathcal{G}_1, \mathcal{G}_2) \geq 0$.*

**Observation 5** *(i) $\text{GED}(\mathcal{G}_1, \mathcal{G}_2) = 0$ iff $\mathcal{G}_1$ is isomorphic to $\mathcal{G}_2$, (ii) $\text{SED}(\mathcal{G}_1, \mathcal{G}_2) = 0$ iff $\mathcal{G}_1$ is subgraph isomorphic to $\mathcal{G}_2$.*

## B  Additional Proofs

### B.1  Proof of Theorem. 1

Our proof relies on two lemmas.

PROOF of Theorem 1. It suffices to prove (i) $\text{SED}(\mathcal{G}_1, \mathcal{G}_2) \geq \widehat{\text{GED}}(\mathcal{G}_1, \mathcal{G}_2)$ and (ii) $\widehat{\text{GED}}(\mathcal{G}_1, \mathcal{G}_2) \geq \text{SED}(\mathcal{G}_1, \mathcal{G}_2)$.

(i) Let $\mathcal{S} = (\mathcal{V}_\mathcal{S}, \mathcal{E}_\mathcal{S}, \mathcal{L}_\mathcal{S}) \subseteq \mathcal{G}_2$ be the subgraph minimizing $\text{SED}(\mathcal{G}_1, \mathcal{S})$ (Recall Eq. 14). Consider the mapping $\pi$ from $\mathcal{G}_1$ to $\mathcal{S}$ corresponding to $\text{SED}(\mathcal{G}_1, \mathcal{S})$ (and hence $\text{GED}(\mathcal{G}_1, \mathcal{S})$ as well). We extend $\pi$ to define a mapping $\widehat{\pi}$ from $\mathcal{G}_1$ to $\mathcal{G}_2$ by mapping of all nodes in set $\mathcal{V}_2 \setminus \mathcal{V}_\mathcal{S}$ to dummy nodes in $\mathcal{G}_1$; the edge mappings are defined analogously.

Under this construction, $\text{SED}(\mathcal{G}_1, \mathcal{S}) = \text{GED}(\mathcal{G}_1, \mathcal{G}_2) =$
$\widehat{\text{GED}}_{\widehat{\pi}}(\mathcal{G}_1, \mathcal{G}_2) \geq \widehat{\text{GED}}(\mathcal{G}_1, \mathcal{G}_2)$. This follows from the property that under $\widehat{d}$, insertion costs are zero, that is $\widehat{d}(\epsilon, \ell) = 0$. Thus, the additional mappings introduced in $\widehat{\pi}$ do not incur additional costs under $\widehat{d}$.

(ii) Consider $\mathcal{S} \subseteq \mathcal{G}_2$ and a mapping $\pi$ from $\mathcal{G}_1$ to $\mathcal{S}$ such that $\text{GED}_\pi(\mathcal{G}_1, \mathcal{S}) = \widehat{\text{GED}}(\mathcal{G}_1, \mathcal{G}_2)$. The existence of such a subgraph is guaranteed (See Lemma 3). From the definition of GED, $\text{GED}_\pi(\mathcal{G}_1, \mathcal{S}) \geq \text{GED}(\mathcal{G}_1, \mathcal{S})$. Furthermore, since $\mathcal{S} \subseteq \mathcal{G}_2$, $\text{GED}(\mathcal{G}_1, S) \geq \text{SED}(\mathcal{G}_1, \mathcal{G}_2)$. Combining all these results, we have $\widehat{\text{GED}}(\mathcal{G}_1, \mathcal{G}_2) \geq \text{GED}(\mathcal{G}_1, S) \geq \text{SED}(\mathcal{G}_1, \mathcal{G}_2)$.
Hence, the claim is proved. □

## B.2 Proof of Thm 2.

PROOF. From Thm. 1, we know $\text{SED}(\mathcal{G}_1, \mathcal{G}_2) = \widehat{\text{GED}}(\mathcal{G}_1, \mathcal{G}_2)$. Combining Obs. 1 with Theorem 1, if we show $\widehat{d}(\ell_1, \ell_3) \leq \widehat{d}(\ell_1, \ell_2) + \widehat{d}(\ell_2, \ell_3)$, then the triangle inequality of $\text{SED}$ is established. We divide the proof into four cases:

**(i)** None of $\ell_1, \ell_2, \ell_3$ is $\epsilon$. Hence, $\widehat{d}(\ell_1, \ell_3) = d(\ell_1, \ell_3)$ and the triangle inequality is satisfied.

**(ii)** $\ell_1 = \epsilon$. The LHS is 0 and hence the triangle inequality is satisfied.

**(iii)** $\ell_1 \neq \epsilon$ and $\ell_2 = \epsilon$. LHS$\leq 1$ and RHS$= 1$. Hence, satisfied.

**(iv)** Only $\ell_3 = \epsilon$. Here, LHS$= 1$ and RHS$\geq 1$.

These four cases cover all possible situations and hence, the triangle inequality is established. $\qquad\square$

## B.3 Proof of Lemma 3

**Lemma 3** *There exists a subgraph $\mathcal{S}$ of $\mathcal{G}_2$ and a node map $\pi$ from $\mathcal{G}_1$ to $\mathcal{S}$ such that $\text{GED}_\pi(\mathcal{G}_1, \mathcal{S}) = \widehat{\text{GED}}(\mathcal{G}_1, \mathcal{G}_2)$.*

PROOF. Let $\pi'$ be a node map from $\mathcal{G}_1$ to $\mathcal{G}_2$ corresponding to $\widehat{\text{GED}}(\mathcal{G}_1, \mathcal{G}_2)$. Let $h_1, \cdots, h_l$ be the nodes of $\mathcal{G}_2$ which are inserted in $\pi'$. Construct subgraph $\mathcal{S}$ of $\mathcal{G}_2$ by removing nodes $h_1, \cdots, h_l$ and their incident edges from $\mathcal{G}_2$. Let $\pi$ be the node map from $\mathcal{G}_1$ to $\mathcal{S}$ which is obtained by removing $h'_1, \cdots, h'_l$ from the domain and $h_1, \cdots, h_l$ from the co-domain of $\pi$. Since insertion costs are 0 in $\widehat{d}$ and $\pi$ contains only non-insert operations, then $\text{GED}_\pi(\mathcal{G}_1, \mathcal{S}) = \widehat{\text{GED}}(\mathcal{G}_1, \mathcal{G}_2)$. Hence, the claim is proved. $\qquad\square$

## B.4 Proof of Lemma 2.

PROOF. Let $\mathbf{x}, \mathbf{y} \in \mathbb{R}^n$ be vectors of dimension $n$. We use the notation $\mathbf{x}[i]$ to denote the $i^{th}$ coordinate of $\mathbf{x}$. We observe that :

$$(\text{ReLU}(\mathbf{x}) + \text{ReLU}(\mathbf{y}))[i] = \text{ReLU}(\mathbf{x})[i] + \text{ReLU}(\mathbf{y})[i] = \text{ReLU}(\mathbf{x}[i]) + \text{ReLU}(\mathbf{y}[i])$$
$$\geq \text{ReLU}(\mathbf{x}[i] + \mathbf{y}[i]) = (\text{ReLU}(\mathbf{x} + \mathbf{y}))[i]$$

Since $\|.\|$ is monotonic, this implies $\|\text{ReLU}(\mathbf{x}) + \text{ReLU}(\mathbf{y})\| \geq \|\text{ReLU}(\mathbf{x} + \mathbf{y})\|$. Using the triangle inequality for $\|.\|$, we get:

$$\|\text{ReLU}(\mathbf{x})\| + \|\text{ReLU}(\mathbf{y})\| \geq \|\text{ReLU}(\mathbf{x}) + \text{ReLU}(\mathbf{y})\| \geq \|\text{ReLU}(\mathbf{x} + \mathbf{y})\| \qquad (15)$$

Substituting $\mathbf{x} = \mathbf{Z}_{\mathcal{G}_\mathcal{Q}} - \mathbf{Z}_{\mathcal{G}_{\mathcal{T}'}}, \mathbf{y} = \mathbf{Z}_{\mathcal{G}_{\mathcal{T}'}} - \mathbf{Z}_{\mathcal{G}_\mathcal{T}}$, we get,

$$\|\text{ReLU}(\mathbf{Z}_{\mathcal{G}_\mathcal{Q}} - \mathbf{Z}_{\mathcal{G}_{\mathcal{T}'}})\| + \|\text{ReLU}(\mathbf{Z}_{\mathcal{G}_{\mathcal{T}'}} - \mathbf{Z}_{\mathcal{G}_\mathcal{T}})\| \geq \|\text{ReLU}(\mathbf{Z}_{\mathcal{G}_\mathcal{Q}} - \mathbf{Z}_{\mathcal{G}_\mathcal{T}})\|.$$

This implies $\mathcal{F}(\mathbf{Z}_{\mathcal{G}_\mathcal{Q}}, \mathbf{Z}_{\mathcal{G}_{\mathcal{T}'}}) + \mathcal{F}(\mathbf{Z}_{\mathcal{G}_{\mathcal{T}'}}, \mathbf{Z}_{\mathcal{G}_\mathcal{T}}) \geq \mathcal{F}_s(\mathbf{Z}_{\mathcal{G}_\mathcal{Q}}, \mathbf{Z}_{\mathcal{G}_\mathcal{T}})$. $\qquad\square$

## C  Complexity Analysis

For this analysis, we make the simplifying assumption that the hidden dimension in the Pre-MLP, GIN and Post-MLP are all $d$. The average density of the graph is $g$. The number of hidden layers in Pre-MLP, and Post-MLP are $L$, and $k$ in GIN.

The computation cost per node for each of these components are as follows.

- **Pre-MLP:** The operations in the MLP involve linear transformation over the input vector $\mathbf{x}_v$ of dimension $|\Sigma|$, followed by non-linearity. This results in $O(|\mathcal{V}|(|\Sigma| \cdot d + d^2 L))$ cost.
- **GIN:** GIN aggregates information from each of the neighbors, which consumes $O(d \cdot g)$ time. The linear transformation consumes an additional $O(d^2)$ time. Applying non-linearity takes $O(d)$ time since it is a linear pass over the hidden dimensions. Finally these operations are repeated over each of the $k$ hidden layers, results in a total $O(k(d^2 + dg))$ computation time per node. Across, all nodes, the total cost is $O(|\mathcal{V}|kd^2 + |\mathcal{E}|kd)$ time. The degree $g$ terms gets absorbed since each edge passes message twice across all nodes.
- **Concatenation:** This step consumes $O(kd)$ time per node.
- **Pool:** Pool iterates over the GIN representation of each node requiring $O(|\mathcal{V}|dk)$ time.
- **Post-MLP:** The final MLP takes $dk$ dimensional vector as input and maps it to a $d$ dimensional vector over $L$ layers. This consumes $O(kd^2 + d^2 L)$ time.

---

**Algorithm 1** BUILDINDEX

---

**Input:** Embeddings $\mathbb{D}$ of graphs
**Output:** Root node of the constructed tree
1: **if** $\mathbb{D} = \emptyset$ **then return** NULL
2: $\mathbf{Z}_\mathcal{P} \leftarrow$ arbitrary embedding in $\mathbb{D}$ as pivot
3: $m_1 \leftarrow \text{MEDIAN}(\{\mathcal{F}_s(\mathbf{Z}_\mathcal{P}, \mathbf{Z}_\mathcal{G}) : \mathbf{Z}_\mathcal{G} \in \mathbb{D} \setminus \{\mathbf{Z}_\mathcal{P}\}\})$
4: $m_2 \leftarrow \text{MEDIAN}(\{\mathcal{F}_s(\mathbf{Z}_\mathcal{G}, \mathbf{Z}_\mathcal{P}) : \mathbf{Z}_\mathcal{G} \in \mathbb{D} \setminus \{\mathbf{Z}_\mathcal{P}\}\})$
5: $\mathbb{D}_1 \leftarrow \{\mathbf{Z}_\mathcal{G} : \mathcal{F}_s(\mathbf{Z}_\mathcal{P}, \mathbf{Z}_\mathcal{G}) \leq m_1, \mathcal{F}_s(\mathbf{Z}_\mathcal{G}, \mathbf{Z}_\mathcal{P}) \leq m_2\}$
6: $\mathbb{D}_2 \leftarrow \{\mathbf{Z}_\mathcal{G} : \mathcal{F}_s(\mathbf{Z}_\mathcal{P}, \mathbf{Z}_\mathcal{G}) \leq m_1, \mathcal{F}_s(\mathbf{Z}_\mathcal{G}, \mathbf{Z}_\mathcal{P}) > m_2\}$
7: $\mathbb{D}_3 \leftarrow \{\mathbf{Z}_\mathcal{G} : \mathcal{F}_s(\mathbf{Z}_\mathcal{P}, \mathbf{Z}_\mathcal{G}) > m_1, \mathcal{F}_s(\mathbf{Z}_\mathcal{G}, \mathbf{Z}_\mathcal{P}) \leq m_2\}$
8: $\mathbb{D}_4 \leftarrow \{\mathbf{Z}_\mathcal{G} : \mathcal{F}_s(\mathbf{Z}_\mathcal{P}, \mathbf{Z}_\mathcal{G}) > m_1, \mathcal{F}_s(\mathbf{Z}_\mathcal{G}, \mathbf{Z}_\mathcal{P}) > m_2\}$
9: **for** $i = 1$ to 4 **do**
10: $\quad t_i \leftarrow \text{BUILDINDEX}(\mathbb{D}_i)$
11: **Return** $\text{NODE}(\mathbf{Z}_\mathcal{P}, m_1, m_2, t_1, t_2, t_3, t_4)$

---

Combining all these factors, the total inference complexity for a graph is $O(|\mathcal{V}|(|\Sigma| \cdot d + d^2 L + kd^2) + |\mathcal{E}|kd)$. This operation is repeated on both the query and target graphs to compute their embeddings, on which distance function $\mathcal{F}$ is operated. Thus, the final cost is $O(n(|\Sigma| \cdot d + d^2 L + kd^2) + mkd)$, where $n = |\mathcal{V}_\mathcal{Q}| + |\mathcal{V}_\mathcal{T}|$ and $m = |\mathcal{E}_\mathcal{Q}| + |\mathcal{E}_\mathcal{T}|$.

## D  Querying in the Embedding Space

We assume the standard querying setup where the database graphs are known apriori, while the query graph is unseen and provided at query time. Since GREED generates pair-independent embeddings, representations of the database graphs can be generated apriori and stored. Furthermore, due to both the predicted GED and SED satisfying the triangle inequality, the embeddings can be indexed. Thus, at query time, we need to perform only two operations: **(1)** embed the query graph $\mathcal{G}_\mathcal{Q}$, and **(2)** scan the database embeddings against the query embedding to compute the answer set. We focus the discussion on the two most common database queries of *range* and *k-NN* queries.

**Definition 5 (Range Query)** *Given a database* $\mathbb{D} = \{\mathbf{Z}_{\mathcal{G}_1}, \cdots, \mathbf{Z}_{\mathcal{G}_n}\}$ *of graph embeddings, a query graph* $\mathcal{G}_\mathcal{Q}$ *and a threshold* $\theta$*, find the answer set* $\mathbb{A} = \{\mathcal{G}_i \mid \mathcal{F}_g(\mathbf{Z}_{\mathcal{G}_i}, \mathbf{Z}_{\mathcal{G}_\mathcal{Q}}) \leq \theta\}$.

**Definition 6 (k-NN Query)** *Given a database* $\mathbb{D} = \{\mathbf{Z}_{\mathcal{G}_1}, \cdots, \mathbf{Z}_{\mathcal{G}_n}\}$ *of graph embeddings, a query graph* $\mathcal{G}_\mathcal{Q}$ *and k, find the database graphs with the k smallest distance to* $\mathcal{G}_\mathcal{Q}$ *as per* $\mathcal{F}_g(\mathbf{Z}_{\mathcal{G}_i}, \mathbf{Z}_{\mathcal{G}_\mathcal{Q}})$.

The above definitions can be adopted for SED by using $\mathcal{F}_s$. For the rest of the discussion, we assume $\mathcal{F}_s$ since due to asymmetry, SED requires some additional considerations.

### D.1  Indexing

We exploit the triangle inequality of GED and SED to index the database embeddings. Alg. 1 presents the pseudocode. We choose a random embedding $\mathbf{Z}_\mathcal{P} \in \mathbb{D}$ as the *pivot* (line 2), based on which we split the remaining embeddings into four groups (lines 3-8). This process continues recursively on each group (lines 9-10) till a partition gets empty (line 1). Note that in GED the distances are

---

**Algorithm 2** RANGEQUERY

---

**Input:** Query embedding $\mathbf{Z}_{\mathcal{G}_\mathcal{Q}}$, threshold $\theta$, root node $t = \text{NODE}(\mathbf{Z}_\mathcal{P}, m_1, m_2, t_1, t_2, t_3, t_4)$
**Output:** $\mathbb{A} \leftarrow \{\mathbf{Z}_\mathcal{G} \mid \mathcal{F}_s(\mathbf{Z}_{\mathcal{G}_\mathcal{Q}}, \mathbf{Z}_\mathcal{G}) \leq \theta\}$
1: **if** $t = $ NULL **then return** $\emptyset$
2: **if** $\mathcal{F}_s(\mathbf{Z}_\mathcal{P}, \mathbf{Z}_{\mathcal{G}_\mathcal{Q}}) \leq m_1 - \theta$ **then**
3: $\quad$ mark $t_3$ and $t_4$ for pruning
4: **if** $\mathcal{F}_s(\mathbf{Z}_{\mathcal{G}_\mathcal{Q}}, \mathbf{Z}_\mathcal{P}) > m_2 + \theta$ **then**
5: $\quad$ mark $t_1$ and $t_3$ for pruning
6: **if** $\mathcal{F}_s(\mathbf{Z}_{\mathcal{G}_\mathcal{Q}}, \mathbf{Z}_\mathcal{P}) \leq \theta - m_1$ **then**
7: $\quad \mathbb{A} \leftarrow \mathbb{A} \cup \mathbb{D}_1 \cup \mathbb{D}_2$
8: $\quad$ mark $t_1$ and $t_2$ for pruning
9: **else**
10: $\quad \mathbb{A} \leftarrow \emptyset$
11: **for** $i = 1$ to 4 **do**
12: $\quad$ **if** $t_i$ is not marked for pruning **then**
13: $\quad\quad \mathbb{A} \leftarrow \mathbb{A} \cup \text{RANGEQUERY}(\mathbf{Z}_{\mathcal{G}_\mathcal{Q}}, \theta, t_i)$
14: **Return** $\mathbb{A}$

---

| Name | $Avg.|\mathcal{V}|$ | $Avg.|\mathcal{E}|$ | $|\Sigma|$ | #Graphs | $Avg.|\mathcal{V}_\mathcal{Q}|$ | $Avg.|\mathcal{E}_\mathcal{Q}|$ |
|---|---|---|---|---|---|---|
| Dblp | $1.66M$ | $7.2M$ | 8 | 1 | 15 | 14 |
| Amazon | $334k$ | $925k$ | 1 | 1 | 12 | 16 |
| PubMed | $19.7k$ | $44.3k$ | 3 | 1 | 12 | 11 |
| CiteSeer | $4.2k$ | $5.3k$ | 6 | 1 | 12 | 12 |
| Cora_ML | $3k$ | $8.2k$ | 7 | 1 | 11 | 11 |
| Protein | 38 | 70 | 3 | 1,071 | 9 | 11 |
| AIDS | 14 | 15 | 38 | 1,811 | 7 | 7 |
| AIDS' | 9 | 9 | 29 | 700 | 9 | 9 |
| Linux | 8 | 7 | 1 | 1,000 | 8 | 7 |
| IMDB | 13 | 65 | 1 | 1,500 | 13 | 65 |

Table A: Datasets

symmetric and hence $m_1$ and $m_2$ will converge. Consequently, we will have two partitions at each node instead of four.

### D.2 Range Query

From the triangle inequality, we can infer the lower bounds listed in lines 2 and 4 of Alg. 2. Hence, if these bounds are larger than $\theta$, the corresponding sub-trees are pruned. Similarly, if the upper bound is smaller than $\theta$ (line 6), the entire sub-tree is added to the answer set (line 7). Otherwise, we recurse (lines 11-13).

### D.3 $k$-NN query

$k$-NN utilizes the same bounds from Alg. 2 to prune and prioritize the search space. However, exploration proceeds in a *best-first search* manner. Alg. 3 presents the pseudocode. Alg. 3 maintains two priority-queues; one to keep track of the $k$-NN till the current stage of the search process ($\mathbb{A}$ in line 1), and the second to store index nodes in ascending order of their lower bound distance ($Cands$ in line 2). We pop the *best* node $\mathcal{P}$ from $Cands$ and include it to the answer set if the distance is within top-$k$ (lines 6-7). Further, the sub-tree at $\mathcal{P}$ is processed if it satisfies the lower bound criteria (lines 8-15). The search ends when either $Cands$ is either empty or the lower bound of the top-most node is larger than the $k$-th distance in $\mathbb{A}$ (line 4). In case of GED, $LB_1$ (line 8) and $LB_2$ (line 9) will converge to the same value due to symmetry .

### D.4 Scaling SED to million-scale graphs

SED is typically encountered in situations where the query is a small graph ($< 50$ nodes) [9, 32] and the target graph is a single large graph, potentially containing millions of nodes and edges. To scale to million-sized target graphs, we perform *neighborhood decomposition*. Specifically, we extract the $k$-hop neighborhood $\mathcal{G}_v$ around each node $v \in \mathcal{V}_\mathcal{T}$ in the target graph $\mathcal{G}_\mathcal{T}$, embed them into feature space using GREED, and then indexed as outlined § D.1. The distance between query $\mathcal{G}_\mathcal{Q}$ and $\mathcal{G}_\mathcal{T}$ is computed as:

---

**Algorithm 3** $k$-NN

**Input:** Query embedding $\mathbf{Z}_{\mathcal{G}_\mathcal{Q}}$, $k$, root node $t = \text{NODE}(\mathbf{Z}_\mathcal{P}, m_1, m_2, t_1, t_2, t_3, t_4)$
**Output:** $\mathbb{A} \leftarrow$ top-$k$ closest embeddings in $\mathbb{D}$
1: $\mathbb{A} \leftarrow$ A priority queue of size up to $k$. Stores entries in descending order of distance.
2: $Cands \leftarrow$ A priority queue. Stores entries in ascending order of distance lower bound.
3: $Cands.\text{insert}\left(\left\langle t, \mathcal{F}_s\left(\mathbf{Z}_{\mathcal{G}_\mathcal{Q}}, \mathbf{Z}_\mathcal{P}\right)\right\rangle\right)$
4: **while** $Cands.size() > 0$ **And** $Cands.top().LB < \mathbb{A}.top().distance$ **do**
5:     $\mathcal{P} \leftarrow Cands.pop()$
6:     **if** $|\mathbb{A}| < k$ **Or** $\mathcal{F}_s\left(\mathbf{Z}_{\mathcal{G}_\mathcal{Q}}, \mathbf{Z}_\mathcal{P}\right) < \mathbb{A}.top().distance$ **then**
7:         $\mathbb{A}.insert\left(\left\langle \mathcal{P}, \mathcal{F}_s\left(\mathbf{Z}_{\mathcal{G}_\mathcal{Q}}, \mathbf{Z}_\mathcal{P}\right)\right\rangle\right)$
8:     $LB_1 \leftarrow |\mathcal{F}_s\left(\mathbf{Z}_\mathcal{P}, \mathbf{Z}_{\mathcal{G}_\mathcal{Q}}\right) - m_1|$
9:     $LB_2 \leftarrow |\mathcal{F}_s\left(\mathbf{Z}_{\mathcal{G}_\mathcal{Q}}, \mathbf{Z}_\mathcal{P}\right) - m_2|$
10:     $LB = \max\{LB_1, LB_2\}$
11:     **if** $LB \leq \theta$ **then**
12:         $Cands.insert(\langle t_1, LB \rangle)$
13:         $Cands.insert(\langle t_2, LB \rangle)$
14:         $Cands.insert(\langle t_3, LB \rangle)$
15:         $Cands.insert(\langle t_4, LB \rangle)$
16: **Return** $\mathbb{A}$

---

$$\mathcal{F}_s\left(\mathcal{G}_\mathcal{Q}, \mathcal{G}_\mathcal{T}\right) = \min_{\forall \mathcal{G}_v \in \mathcal{G}_\mathcal{T}} \left\{\mathcal{F}_s\left(\mathcal{G}_\mathcal{Q}, \mathcal{G}_v\right)\right\} \tag{16}$$

We next show that as long as the $k$-hop neighborhoods are *sufficiently large*, the proposed neighborhood decomposition strategy is optimal.

**Theorem 3** *A nearest subgraph $\mathcal{S} \subseteq \mathcal{G}_\mathcal{T}$ to $\mathcal{G}_\mathcal{Q}$ can be found in an $l/2$-hop neighborhood $\mathcal{G}_v \subseteq \mathcal{G}_\mathcal{T}$, centered at some $v \in \mathcal{V}_{\mathcal{G}_\mathcal{T}}$, where $l$ is the length of the longest path in $\mathcal{G}_\mathcal{Q}$.*

PROOF. From Thm. 1, $\text{SED}(\mathcal{G}_\mathcal{Q}, \mathcal{G}_\mathcal{T}) = \widehat{\text{GED}}(\mathcal{G}_\mathcal{Q}, \mathcal{G}_\mathcal{T})$. Let $\pi$ be the node map produced by $\widehat{\text{GED}}(\mathcal{G}_\mathcal{Q}, \mathcal{G}_\mathcal{T})$. Let us now consider the subgraph $\mathcal{S} \subseteq \mathcal{G}_\mathcal{T}$ induced by all node maps of $\pi$ that are not inserts. It follows that $\text{SED}(\mathcal{G}_\mathcal{Q}, \mathcal{G}_\mathcal{T}) = \widehat{\text{GED}}(\mathcal{G}_\mathcal{Q}, \mathcal{G}_\mathcal{T}) = \text{SED}(\mathcal{G}_\mathcal{Q}, \mathcal{S})$. Since there are no inserts, the topology of $\mathcal{S}$ is a subgraph of $\mathcal{G}_1$ (i.e., subgraph isomorphic if we ignore label information). Hence, the diameter of $\mathcal{S}$ is $\leq$ the length $l$ of the longest path in $\mathcal{G}_\mathcal{Q}$. Since $\mathcal{S} \subseteq \mathcal{G}_\mathcal{T}$, $\mathcal{S}$ is contained in some $l/2$ neighborhood $\mathcal{G}_v$ of $\mathcal{G}_\mathcal{T}$. □

Finding the length of the longest path is NP-hard. However, we do not need to find the exact length of the longest path: any upper bound suffices. Moreover we do not even need the length of the longest path for $l$. The nearest subgraph having a diameter equal to $l$ is rare. In practice, the diameter of the nearest subgraph is unlikely to be much higher than the diameter of the query itself.

## E   Datasets

`Dblp:` `Dblp` is a co-authorship network where each node is an author and two authors are connected by an edge if they have co-authored a paper. The label of a node is the venue where the authors has published most frequently. The dataset has been obtained from https://www.aminer.org/citation.
`Amazon` [28]**:** Each node in `Amazon` represents a product and two nodes are connected by an edge if they are frequently co-purchased. The graph is unlabeled and hence equivalent to a graph containing a single label on all nodes.
`PubMed:` `PubMed` dataset is a citation network which consists of scientific publications from `PubMed` database pertaining to diabetes classified into one of three classes.
`Protein:` `Protein` dataset consists of protein graphs. Each node is labeled with a one of three functional roles of the protein.
`AIDS:` `AIDS` dataset consists of graphs constructed from the AIDS antiviral screen database. These graphs representing molecular compounds with Hydrogen atoms omitted. Atoms are represented as nodes and chemical bonds as edges.
`AIDS':` `AIDS'` dataset is another collection of graphs constructed from the AIDS antiviral screen database. The graphs and their properties differ from those in `AIDS`. These graphs also represent chemical compound structures.
`CiteSeer:` `CiteSeer` is a citation network which consists of scientific publications classified into one of six classes. Generally a smaller version is used for this dataset, but we use the larger version from [8].
`Cora_ML:` Cora dataset is a citation dataset consisting of many scientific publications classified into one of seven classes based on paper topic. `Cora_ML` is a smaller datset extracted from Cora [8].
`Linux:` `Linux` dataset is a collection of program dependence graphs, where each graph is a function and nodes represent statements while edges represent dependency between statements.
`IMDB:` IMDB dataset is a collection of ego-networks of actors/actresses that have appeared together in any movie.

## F   Baselines

GMN explicitly assumes the distance function to be symmetric, which violates SED. GRAPHSIM has an assumption that a large difference in size of the query and target graphs leads to a large distance, which is not true in SED. In GOTSIM, there is an explicit assumption of modeling a symmetric distance function since they use cosine similarity to compare node neighborhoods The normalization factor in Eq. 6 of [14] is also based on whole graph matching. Finally, GENN-A* is not included for SED since it does not scale on graphs beyond 10 nodes (See § 4.3).

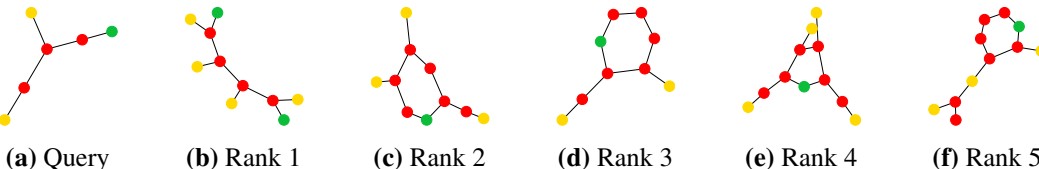

| **(a)** Query | **(b)** Rank 1 | **(c)** Rank 2 | **(d)** Rank 3 | **(e)** Rank 4 | **(f)** Rank 5 |

Figure E: **Visualizations of query and resulting matches produced by GREED. Red, Green and Yellow colors indicate Carbon, Nitrogen and Oxygen atoms respectively. The actual and predicted SED for the target graphs are (b)** $0, 0.4$, **(c)** $0, 0.5$, **(d)** $1, 0.6$, **(e)** $0, 0.6$ **and (f)** $1, 0.6$**.**

## G  Ablation Study

**Impact of GIN:** To highlight the importance of GIN, we conduct ablation studies by replacing the GIN convolution layers in the model with several other convolution layers. As visible in the Table B, GIN consistently achieves the best accuracy. This is not surprising since GIN is provably the most expressive among GNNs in distinguishing graph structures (essential to SED or GED computation) and is as powerful as the Weisfeiler-Lehman Graph Isomorphism test [47].

| Methods | CiteSeer (SED) | PubMed (SED) | Amazon (SED) | IMDB (GED) |
|---|---|---|---|---|
| **GREED (GIN)** | **0.519** | **0.728** | **0.495** | **6.734** |
| **GREED-GCN** | 0.556 | 0.756 | 0.532 | 12.151 |
| **GREED-GraphSage** | 1.364 | 1.156 | 1.841 | 91.312 |
| **GREED-GAT** | 1.294 | 1.259 | 1.843 | 89.034 |

Table B: **Ablation studies: GIN vs others. RMSE produced by different methods are shown and GREED with GIN produces the best results.**

| Pool functions | CiteSeer (SED) | PubMed (SED) | Amazon (SED) | IMDB (GED) |
|---|---|---|---|---|
| **GREED (Sum)** | **0.519** | 0.728 | **0.495** | **6.734** |
| **GREED-Max** | 0.795 | **0.709** | 0.603 | 52.519 |
| **GREED-Mean** | 0.922 | 0.732 | 0.846 | 52.483 |
| **GREED-Attention** | 0.914 | 0.797 | 0.868 | 130.47 |

Table C: **Ablation studies: sum-pool vs others. The sum-pool is the best choice among the considered alternatives.**

**Impact of sum-pool:** To substantiate our choice of the pooling layer, we have performed ablation studies with various pooling functions as replacements for sum-pool. It is clear from Table C that sum-pool is the best choice among the considered alternatives.

Sum-pool can better distinguish graph sizes better than other aggregation functions such as mean-pool or max-pool. To elaborate, let us consider a graph $G_1$ that is significantly larger than another graph $G_2$. In this scenario, the individual coordinates of $G_1$'s embedding can potentially be significantly larger than those of $G_2$ since in $G_1$ the summation is being done over a larger set of embeddings. Both mean-pool and max-pool fail to capture the size information as effectively, since the max and the mean operations do not scale with the number of inputs.

**Impact of Pre-mlp layer:** Table D presents the results. As visible, we do not see any significant difference in performance on average.

**Performance of GREED-NN and GREED-Dual on GED**: Fig. F presents the results. The trends are similar to what we observed for SED in Sec. 4.5. GREED-NN closes the performance gap with GREED as more training data is provided. Furthermore, GREED-dual is consistently worse that GREED.

## H  Alignment

In real-world applications of subgraph similarity search, alignments are of interest only for a small number of similar subgraphs. Our framework is intended to serve as a filter to retrieve this small set of similar subgraphs from a large number of candidates. To elaborate, a graph database may contain

| RMSE | With Pre-MLP | Without Pre-MLP |
|---|---|---|
| AIDS' (SED) | 0.51 | 0.51 |
| Amazon | 0.5 | **0.39** |
| CiteSeer | 0.52 | **0.51** |
| Cora_ML | **0.64** | 0.68 |
| AIDS (GED) | **0.8** | 0.85 |
| IMDB (GED) | **6.73** | 7.68 |
| Linux (GED) | 0.42 | **0.41** |
| Protein | 0.52 | 0.52 |
| PubMed | 0.73 | 0.73 |

Table D: RMSE of GREED with and without the Pre-mlp layer.

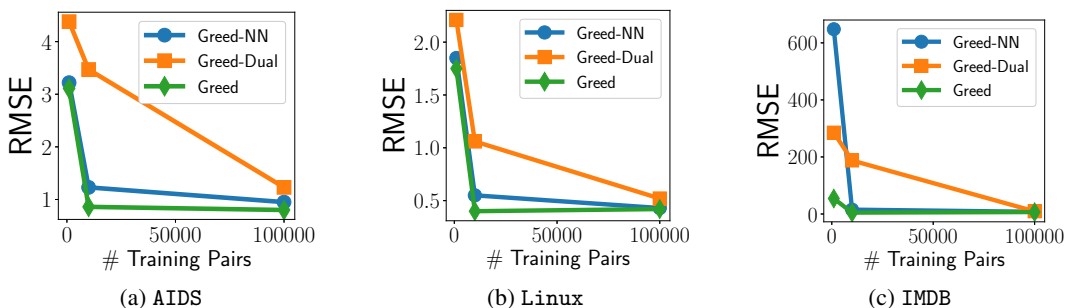

(a) AIDS  (b) Linux  (c) IMDB

Figure F: Performance of GREED-NN and GREED-Dual on GED. Refer to § 4.5 for details.

| Methods | PubMed | Amazon |
|---|---|---|
| **GREED Retrieval** | 0.373 | 6.471 |
| **MIP-F2 Alignment** | 52.8 | 68.4 |

Table E: **The average running times in seconds per top-10 query. Our technique is much faster than MIP-F2 alignment.**

thousands or millions of graphs (or alternatively, thousands or millions of neighborhoods of a large graph) which need to be inspected for similar subgraphs. A user is typically interested in only a handful of these subgraphs that are highly similar to the query. Since the filtered set is significantly smaller, a non-neural exact algorithm suffices to construct the alignments (Lemma 1 allows us to adapt general cost GED alignment techniques for SED alignment). Computing alignments across the entire database is unnecessary and slows down the query response time.

To substantiate our claim, we show the average running time for answering 10-NN queries. We break up the running time into two components: (i) 10-NN retrieval time by GREED, (ii) exact alignment time using MIP-F2 for the 10-NN neighborhoods retrieved by GREED. We observe that exact alignment by existing methods on the 10-NN neighborhoods completes in reasonable time. In contrast, GENN-A$^*$ does not scale on either PubMed or Amazon since it computes alignments across *all* (sub)graphs.

## I Visualization

Searching for molecular fragment containment is a routine task in drug discovery [20]. Motivated by this, we show the top-5 matches to an SED query on the AIDS dataset produced by GREED in Fig. E. The query is a functional group (Hydrogen atoms are not represented). GREED is able to extract chemical compounds that contain this molecular fragment (except for ranks 3 and 5, which contain this group with 1 edit) from around 2000 chemical compounds with varying sizes and structures. This validates the efficacy of GREED at a semantic level.

## J Subgraph sampling strategies for SED generalizability

In RWR, we perform fixed length random walks, where the length of a walk is the average diameter size of the queries generated through BFS during training. Next, we merge the walks to form a graph.

In Rw, we perform random walks, till the diameter of the resultant graph is the same as the average diameter of the BFS sampled train graphs.

The details of the SHADOW sampler is explained in [48].

## K  Extension to maximum common subgraph similarity (MCSS)

Besides GED, MCSS is also a popular similarity measure for whole-graph comparison. Except the inductive bias injected through $\mathcal{F}$, all components of GREED is generic for any graph distance function. In this section, we replace $\mathcal{F}$ with an MLP and model MCSS. Table F presents the results. As visible, the trends hold even in this distance function, where GREED outperforms the closest baseline of $H^2MN$.

| Methods | AIDS | Linux | IMDB |
|---------|------|-------|------|
| GREED | **0.514** | **0.085** | **0.293** |
| $H^2MN$ | 0.652 | 0.152 | 0.475 |

Table F: **RMSE on MCSS.**

## L  Heat Maps for Prediction Error

In Figures G to P, we show the variation of the errors on SED and GED prediction with query sizes and ground truth values for GREED and the baselines on all the corresponding datasets. This experiment is an extension of the heat-map results in Fig. 3 in the main paper. These datasets show variations in the distributions of SED and GED values. It is interesting to observe that among the baselines, different methods perform well on different regions (i.e., combinations of high/low SED and high/low query sizes). The baselines do not show good performance on all regions. However, for GREED, we see a much better coverage for all types of regions in the domain. Furthermore, for every region, our models outperform (or are at least competitive with) the best performing baseline for that region.

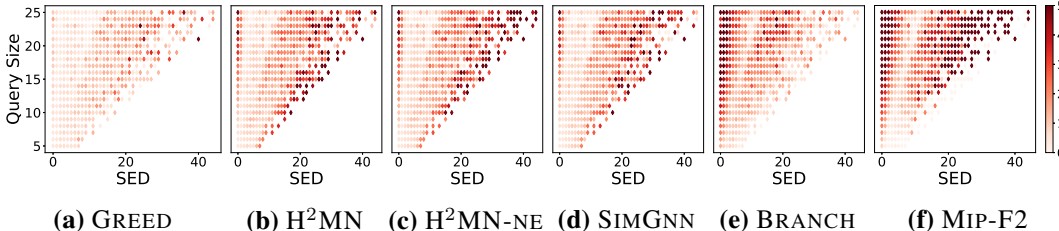

(a) GREED    (b) $H^2MN$    (c) $H^2MN$-NE    (d) SIMGNN    (e) BRANCH    (f) MIP-F2

Figure G: **Heat Maps of SED error against query size and SED values for** `Dblp`. **Darker means higher error.**

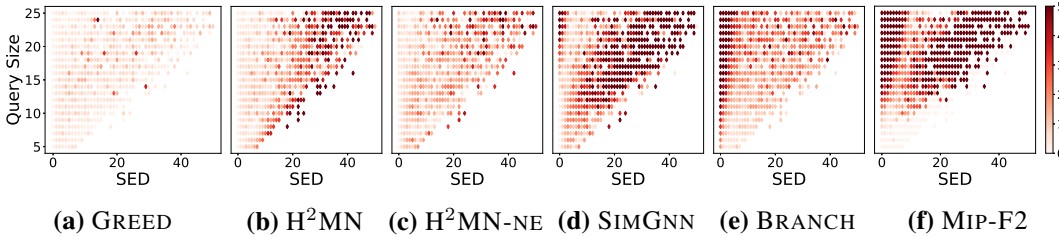

(a) GREED    (b) $H^2MN$    (c) $H^2MN$-NE    (d) SIMGNN    (e) BRANCH    (f) MIP-F2

Figure H: **Heat Maps of SED error against query size and SED values for** `Amazon`. **Darker means higher error.**

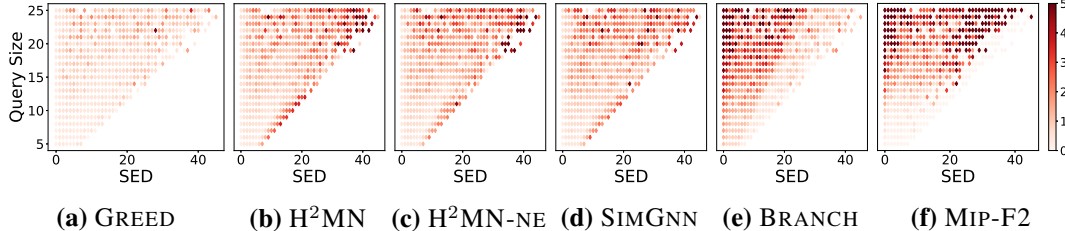

**(a)** GREED    **(b)** H²MN    **(c)** H²MN-NE    **(d)** SIMGNN    **(e)** BRANCH    **(f)** MIP-F2

Figure I: **Heat Maps of SED error against query size and SED values for** `PubMed`**. Darker means higher error.**

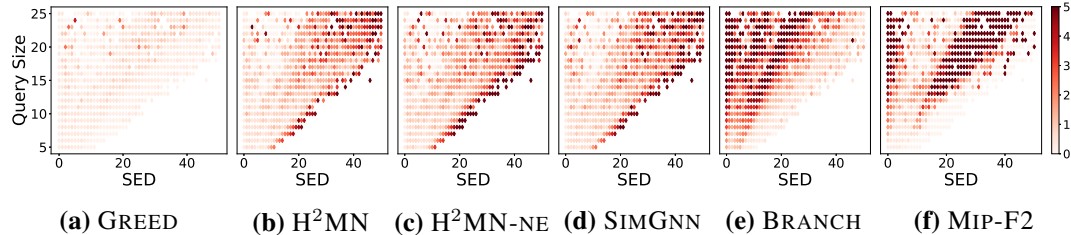

**(a)** GREED    **(b)** H²MN    **(c)** H²MN-NE    **(d)** SIMGNN    **(e)** BRANCH    **(f)** MIP-F2

Figure J: **Heat Maps of SED error against query size and SED values for** `CiteSeer`**. Darker means higher error.**

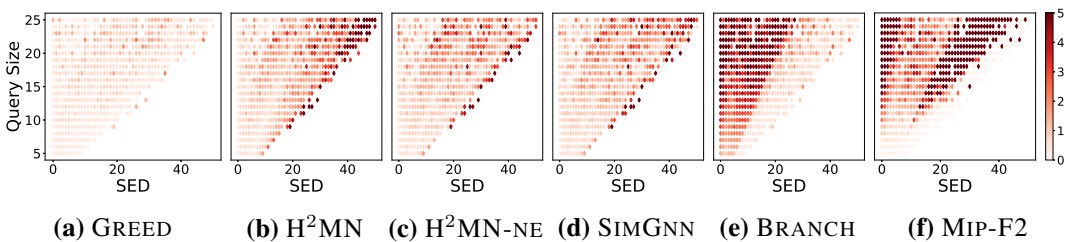

**(a)** GREED    **(b)** H²MN    **(c)** H²MN-NE    **(d)** SIMGNN    **(e)** BRANCH    **(f)** MIP-F2

Figure K: **Heat Maps of SED error against query size and SED values for** `Cora_ML`**. Darker means higher error.**

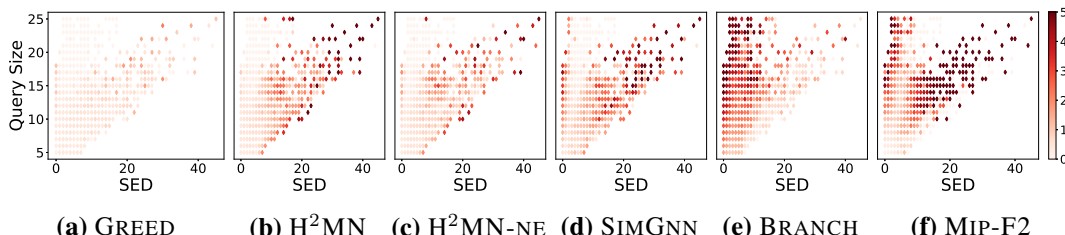

**(a)** GREED    **(b)** H²MN    **(c)** H²MN-NE    **(d)** SIMGNN    **(e)** BRANCH    **(f)** MIP-F2

Figure L: **Heat Maps of SED error against query size and SED values for** `Protein`**. Darker means higher error.**

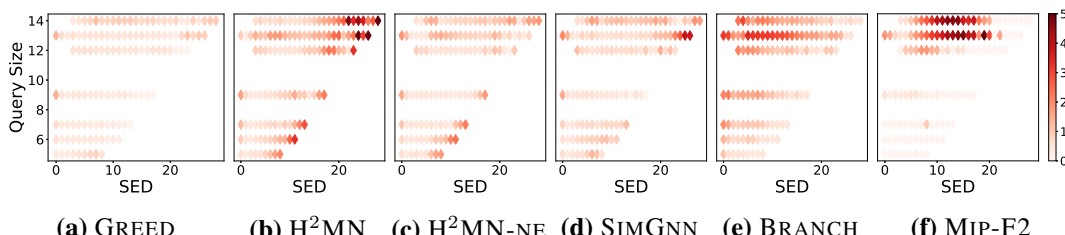

**(a)** GREED    **(b)** H²MN    **(c)** H²MN-NE    **(d)** SIMGNN    **(e)** BRANCH    **(f)** MIP-F2

Figure M: **Heat Maps of SED error against query size and SED values for** `AIDS`**. Darker means higher error.**

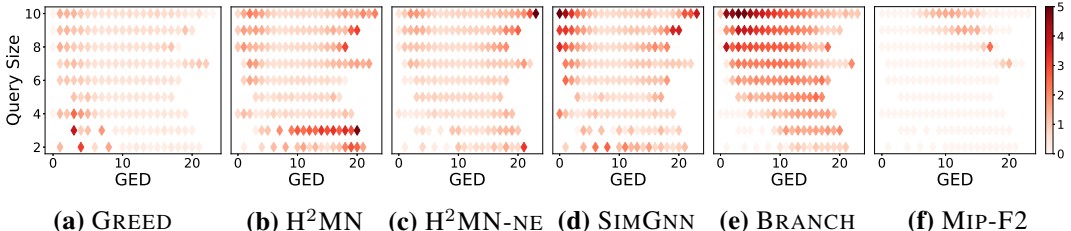

**(a)** GREED    **(b)** H²MN    **(c)** H²MN-NE    **(d)** SIMGNN    **(e)** BRANCH    **(f)** MIP-F2

Figure N: **Heat Maps of GED error against query size and GED values for** `AIDS'`**. Darker means higher error.**

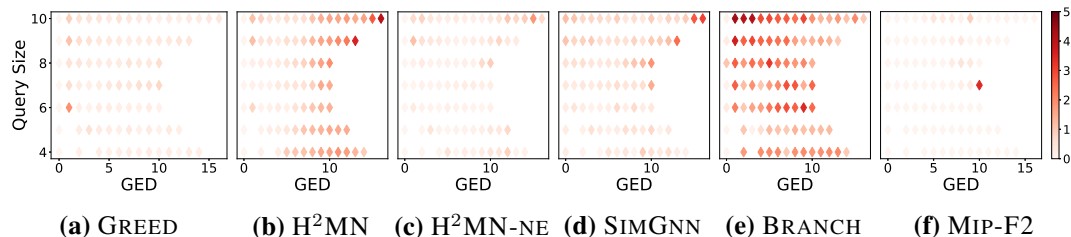

**(a)** GREED    **(b)** H²MN    **(c)** H²MN-NE    **(d)** SIMGNN    **(e)** BRANCH    **(f)** MIP-F2

Figure O: **Heat Maps of GED error against query size and GED values for** `Linux`**. Darker means higher error.**

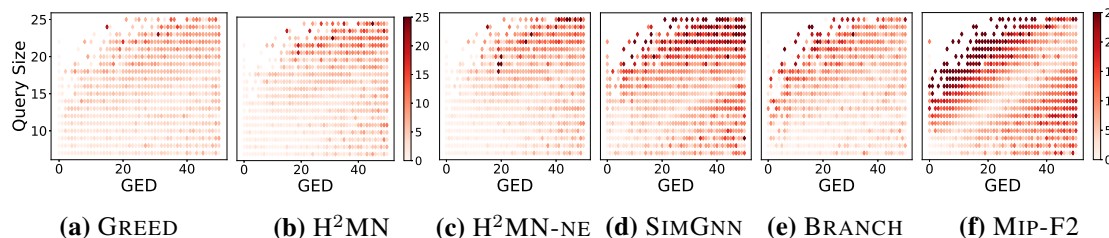

**(a)** GREED    **(b)** H²MN    **(c)** H²MN-NE    **(d)** SIMGNN    **(e)** BRANCH    **(f)** MIP-F2

Figure P: **Heat Maps of GED error against query size and GED values for** `IMDB`**. Darker means higher error.**