# OpenReview forum: "GREED: A Neural Framework for Learning Graph Distance Functions"
_NeurIPS.cc/2022/Conference — NeurIPS 2022 Accept_

### Official Review · Reviewer_ebHE · 2022-07-11

**Rating:** 6
**Confidence:** 5
**Soundness:** 3 good
**Presentation:** 2 fair
**Contribution:** 2 fair

**Summary:**

In this work, the authors propose a neural framework to learn two well established metric, GED and SED, for graph similarity computation. By designing a siamese graph neural network to learn GED and SED in a property-preserving manner, the authors also show that the learned functions can guarantee to be a distance metric. Experiments demonstrate the effectiveness of the proposed framework in terms of accuracy and efficiency.

**Questions:**

Besides the above weakness, one more question is from Section 4.5:

From Figure i-k, we can see that Greed-NN shows better performance compared Greed in two out of three datasets, i.e., on PubMed and CiteSeer. The authors should give some analyze of this phenomenon, which is very important to justify their claimed inductive bias.


**Strengths And Weaknesses:**

### Strengths
1. The idea to unify GED and SED in one learning manner is kind of interesting.

2. The proposed framework is reasonable to approximate GED and SED.

3. The experiments are comprehensive.

### Weaknesses
1. There have been many works, as mentioned by the authors, on learning graph-based similarity metrics by GNN, thus in general, the novelty of this idea is limited. It is straightforward to include SED into the framework.

2. Though useful, the proposed GNN framework is not novel, which have been proposed in previous works. Especially, the motivation to use Siamese GNN is not clear.

3. The presentation of this work need improvements:

    a) The title ``learning graph distance functions’’ is kind of overclaimed, since the work only learns for graph edit distance, while there are other distances, like maximum common subgraph.

    b) The content of Section 2 is a bit too much, since most of them are established knowledge, not proposed by the authors. The authors can shorten Section 2 by moving some materials to appendix. Moreover, it would be better the authors give the references for GED, SED, and the properties.

    c) In Section 4, the ref of some tables are wrong, e.g., the text following ``4.2 Prediction Accuracy of SED and GED`` and ``4.3 Efficiency`` are all Table 2a and 2b, which is confusing

    d) In Section 4.6, the authors mentioned Table 3a is not found in the paper.

    e) In the Readme of the code, the submission information is still KDD 2022. In fact, I think this is the reason that the table references are messy in the paper. The authors should carefully proofread the paper for NeurIPS submission.

---

> ### Author Response · Authors · 2022-08-02
> **Response to Reviewer ebHE**
>
> **There have been many works, as mentioned by the authors, on learning graph-based similarity metrics by GNN, thus in general, the novelty of this idea is limited. It is straightforward to include SED into the framework.**
>
> *Response:* We do not claim novelty on the problem formulation. Our claim of novelty is on:
> * **Massive scalability:** We are 1000 times faster than the closest baseline of H2MN (Sec 4.4).
> * **Novel Architecture:** The proposed architecture has two key innovations. (1) The embeddings are computed in a pair-independent manner. This is possible only because we use a siamese architecture. (2) We preserve properties of the original space, i.e., the predicted GED space is metric and the predicted SED space satisfies triangle inequality. None of the existing techniques do that. Owing to these properties, the graph embeddings are indexable and this leads to the massive scalability mentioned above.
> * **Better quality:** The proposed algorithm is consistently better than the baselines across most scenarios and scales better with graph sizes.
>
> **Though useful, the proposed GNN framework is not novel, which have been proposed in previous works. Especially, the motivation to use Siamese GNN is not clear.**
>
> *Response:* The first para of Sec 3.2 discusses why we use a siamese architecture. We reproduce it verbatim below.
>
> > If we do not use a siamese architecture, then the embedding model for the query and the target graphs would be different. Hence, the predicted distance would violate symmetry, and therefore, would not be a metric. Furthermore, if the distance computations are pair-dependent [27, 41, 2], i.e., it jointly learns the embedding of the query and the target, then a single graph may correspond to multiple representations. Hence, it would not be a metric or satisfy the triangle inequality.
>
> In addition, we also refer to Sec 4.5, where we compare GREED with its non-siamese version, GREED-dual, where the weights are not shared. As visible in Fig. 4i-4k, GREED is consistently better than GREED-dual.
>
> Regarding the statement "the proposed GNN framework is not novel, which have been proposed in previous works.", we would appreciate it if the reviewer can point out which prior work has proposed this framework.
>
> **The presentation of this work need improvements:**
>
> **a) The title "learning graph distance functions" is kind of overclaimed, since the work only learns for graph edit distance, while there are other distances, like maximum common subgraph.**
>
> *Response:*  We acknowledge this feedback.
>
> * We would be happy to change our title to "GREED: A Neural Framework for Learning Graph Edit Distances" if the reviewer feels so. The acronym GREED actually expands to only edit distances (GRaph Embeddings for Edit Distances). In the title, we followed the naming convention of the baselines such as H2MN[42] and GENN-A*[37], which used the terms "graph similarity learning" or "graph distance learning", despite focusing on graph edit distance.
>
> * Nonetheless, to show generalizability to maximum common subgraph similarity (MCSS), we have added this experiment in Appendix K with reference from Sec 5 in main paper. The same results are also produced below.
>
> RMSE on MCSS:
> | Method |Aids'| Linux |  IMDB |
> |--| -------- | -------- | ---- |
> |Greed |0.514 |0.085 | 0.293    |
> |H2MN |  0.612   |0.152 | 0.475 |
>
>
>
> **b) The content of Section 2 is a bit too much, since most of them are established knowledge, not proposed by the authors. The authors can shorten Section 2 by moving some materials to appendix. Moreover, it would be better if the authors give the references for GED, SED, and the properties.**
>
> *Response:* We acknowledge this feedback. Section 2 has been modified accordingly.
>
> **c) In Section 4, the ref of some tables are wrong, e.g., the text following 4.2 Prediction Accuracy of SED and GED and 4.3 Efficiency are all Table 2a and 2b, which is confusing.**
>
> *Response:* We apologize for the latex reference error. We have corrected them. Now, Tables 1a and 1b present the RMSE errors of the algorithms, and Tables 3a-c present their running-time results.
>
> **d) In Section 4.6, the authors mentioned Table 3a is not found in the paper.**
>
> *Response:* We apologize again for this mistake, which was due to latex reference errors. We have corrected this. Table 3 corresponds to Table 4 in the revised version.
>
> **e) In the Readme of the code, the submission information is still KDD 2022. In fact, I think this is the reason that the table references are messy in the paper. The authors should carefully proofread the paper for NeurIPS submission.**
>
> *Response:* The table references messed up due to latex reference errors. We have also updated the README now.

---

> > ### Author Response · Authors · 2022-08-02
> > **Appeal to the reviewer**
> >
> > **Appeal to the reviewer:** We have addressed all the presentation issues in our rebuttal. We have also clarified why a Siamese network is necessary and the resulting consequence of better quality than all baselines, massive scalability advantage and indexable embeddings. If the reviewer feels the revised version addresses the concerns raised, we humbly request to please raise the rating.

---

> > > ### Author Response · Authors · 2022-08-06
> > > **Looking forward to your post-rebuttal feedback**
> > >
> > > Thanks again for your insightful suggestions and comments. As the deadline for discussion is approaching, we are glad to provide any additional clarifications that you may need.
> > >
> > > In our previous response, we have carefully studied your comments and addressed the concerns raised. We summarize our responses with regards to the following aspects:
> > >
> > > 1. We have addressed all the issues related to presentation. The changes are highlighted in blue in the revised version.
> > >
> > > 2. We explained why the siamese architecture is required for the theoretical guarantees we provide with respect to preservation of distance properties and its implication on the accuracy with actual empirical data.
> > >
> > > 3. We provided preliminary evidence on the efficacy of our technique to MCSS. Nonetheless, we would also be ok to change our title to only edit distance as suggested.
> > >
> > > We hope that the provided new experiments and additional explanations have convinced you of the merits of our work. Please do not hesitate to contact us if there are other clarifications or experiments we can offer.
> > >
> > > Thank you for your time again!
> > >
> > > Best,
> > >
> > > Authors

---

> > ### Comment · Reviewer_ebHE · 2022-08-07
> > **Still have one major concern.**
> >
> > Hi authors,
> >
> > Thanks for your comprehensive responses, especially the writing of the paper, which addresses some of my concerns
> >
> > However, my major concern is that the contributions of this work including idea and framework is not that big, since
> >
> > 1. I agree that the efficiency improvement of the proposed framework is quite impressive. However, in my opinion, the novelty of learning graph edit distance by GNN is limited. My concern is that it might not be enough to only have the efficiency improvement as a NeurIPS submission.
> >
> > 2. In terms of the novelty of the architecture, the novelty lies in the use of siamese architecture, which are two GNNs of share parameters. While the architecture of GNNs ( including Pre-MLP, GIN, Concatenation, Pool, and Post-MLP, and GED/SED prediction), are standard in the literature. This is what I mean `The architecture is not novel in the review`.
> >
> > 3. In terms of the ablation studies of the siamese architecture, your claim `As visible in Fig. 4i-4k, GREED is consistently better than GREED-dual.` is not accurate for me. For example, in 4(I), the performance of Greed-Dual and Greed are nearly the same. Besides, can you give any analysis why Greed-NN outperforms Greed-Dual in most cases? Considering GNN, despite not using siamese architecture, should have better modeling ability than MLP due to the inductive bias of approximating GED/SED?
> >
> > In summary, due to the above concern, I think this paper is borderline paper. Based on the updated version, especially the writings of the paper, I am inclined to borderline accept.

---

> > > ### Author Response · Authors · 2022-08-07
> > > **Clarification on the concerns: Part1**
> > >
> > > Thank you again for your time on reviewing our work and your willingness to raise the rating to borderline accept. (The rating still reflects 4, which corresponds to borderline reject. We would be thankful if you raise it to 5, which corresponds to borderline accept).
> > >
> > > Please find below our clarifications on your queries.
> > >
> > > **I agree that the efficiency improvement of the proposed framework is quite impressive. However, in my opinion, the novelty of learning graph edit distance by GNN is limited. My concern is that it might not be enough to only have the efficiency improvement as a NeurIPS submission.**
> > >
> > > Thank you for your positive comment on the efficiency aspect. We also note that our accuracy is better than state-of-the-art algorithms as well, generalizes to both GED and SED, preserves properties from their original distance spaces, and scales better with query graph sizes.
> > >
> > > **In terms of the novelty of the architecture, the novelty lies in the use of siamese architecture, which are two GNNs of share parameters. While the architecture of GNNs (including Pre-MLP, GIN, Concatenation, Pool, and Post-MLP, and GED/SED prediction), are standard in the literature. This is what I mean The architecture is not novel in the review.**
> > >
> > > We agree with the reviewer that the individual components of the GNN used as the embedding layer are not novel and have been used in the literature. In our humble opinion, the novelty lies in our custom prediction functions (Sections 3.1.1 and 3.1.2) that allow us to guarantee preservation of the metric property for GED predictions and the triangle inequality (along with non-negativity) for SED predictions. These properties are required for many higher order applications, such as clustering and indexing, and GREED is therefore the only algorithm that is suitable for such scenarios. While the prediction function is simply the Euclidean distance on the embedding space for GED, a suitable function for SED is significantly less obvious, and along with corresponding analytical proofs, is one of the novel contributions of our work.
> > >
> > > Furthermore, the Siamese architecture is crucial for these guarantees as it constrains the model to learn a single map from the graph space to the embedding space for both the query and target graphs. The noteworthy achievement of our work is to demonstrate that we can satisfy all these desiderata while still out-performing with large margins, both in terms of accuracy and runtime, state-of-the-art models that freely use pair-dependent cross-graph information in expensive computations, and do not provide any guarantees. We reflect that this is possible because both the Siamese architecture and the custom prediction functions, apart from enabling the requisite guarantees, serve as strong inductive biases for graph distance learning tasks.  As such, we humbly consider the simplicity of our architecture as a salient feature.

---

> > > > ### Author Response · Authors · 2022-08-07
> > > > **Clarification on the concerns: Part2**
> > > >
> > > > **In terms of the ablation studies of the siamese architecture, your claim As visible in Fig. 4i-4k, GREED is consistently better than GREED-dual. is not accurate for me. For example, in 4(I), the performance of Greed-Dual and Greed are nearly the same. Besides, can you give any analysis why Greed-NN outperforms Greed-Dual in most cases? Considering GNN, despite not using siamese architecture, should have better modeling ability than MLP due to the inductive bias of approximating GED/SED?**
> > > >
> > > > The referred statement is an oversight on our part. We have changed it to:
> > > >
> > > > > The RMSE of Greed is *generally* better than GREED-Dual, with the difference being more significant at low volumes.
> > > >
> > > > Greed-NN Vs. Greed-Dual:
> > > >
> > > > We stress that in these ablations, we wish to demonstrate the efficacy of our inductive biases with low volumes of training data. This ability is important since generating training data itself is NP-hard (due to computing GED/SED). To provide concrete examples, with 1000 training pairs, Greed consistently outperforms both Greed-NN and Greed-Dual. With larger training pairs, the effect of inductive bias becomes less important in some of our datasets, and we find that occasionally Greed-NN or Greed-Dual gives better accuracy. However, even in these cases the difference is only marginal, and taking the runtime and theoretical guarantees of Greed into consideration, it would still be the recommended choice for most use cases. Finally, we note that Greed-NN and Greed-Dual have strictly more parameters than Greed, which works in their favour only when the training data is large.
> > > >
> > > > As for why Greed-NN might be better than Greed-Dual, we would like to offer the following insight. Greed-NN keeps the inductive bias due to the Siamese architecture but ablates the inductive bias of the custom prediction function. On the other hand, Greed-Dual keeps the inductive bias due to the custom prediction function but ablates the inductive bias due to the Siamese architecture. A priori, it is difficult to say which of the two sources of inductive biases is more important. A posteriori, however, the observed phenomenon of Greed-NN generally out-performing Greed-Dual can be interpreted as evidence for the Siamese architecture providing a stronger inductive bias than the custom prediction function. Note that the modeling ability of GNN vs MLP should not be used as proxy for the modeling ability of Greed-Dual vs Greed-NN as GNN and MLP perform very different functions, namely graph embedding and GED/SED prediction respectively, in these architectures.
> > > >
> > > > We have added the above discussion to the new revised version.
> > > >
> > > > We once again thank the reviewer for engaging with us in a discussion and helping us improve the paper.

---

> > > > > ### Comment · Reviewer_ebHE · 2022-08-08
> > > > > **Thanks for your clarification.**
> > > > >
> > > > > Based on your latest responses, my major concerns are addressed.
> > > > >
> > > > > I would like to improve my ratings to weak accept.

---

### Official Review · Reviewer_ap6M · 2022-07-12

**Rating:** 6
**Confidence:** 5
**Soundness:** 3 good
**Presentation:** 3 good
**Contribution:** 2 fair

**Summary:**

In this work, the authors design a novel siamese graph neural network GREED which can learn graph (GED) and subgraph edit distances (SED) in a property-preserving manner. The authors demonstrate the efficacy of their approach via empirical results.

**Questions:**

q1) Why did you not cite and compare your approach against the following state-of-the-art recent approaches i.e., GOTSim [3], NeuroMatch [4], Graph embedding network (GEN) [5], and IsoNet [6] ?

q2) Did you try any other alternatives to siamese graph neural network settings i.e., non-sharing of parameters between query and target graphs to see if it helps ?

q3) Did you try any alternatives to the Graph Isomorphism Network (GIN) module ?

q4) Why did you use sum pool specifically ?

**Limitations:**

The authors list just one limitation of their work.

**Strengths And Weaknesses:**

Here are the main strengths of the current work :
1) The authors focus on graph similarity measurement which is a very important problem in the graph ML community.
2) The authors through empirical results demonstrate that the proposed approach works well as well as is significantly faster compared to the other baseline approaches.
3) The authors via using a siamese graph neural network which utilizes shared parameters to embed both the query and target graphs respectively are able to reduce the model parameters as well as the training time significantly.

Here are the main weaknesses of the current work :
1) The proposed approach has limited novelty factor.
2) The authors do not seem to compare their work with state-of-the-art recent approaches which could have been good baselines to compare against. More concretely, the authors do not cite and/or compare against the following state-of-the-art recent approaches i.e., GOTSim [3], NeuroMatch [4], Graph embedding network (GEN) [5], and IsoNet [6].
3) Some aspects of the paper lack details/discussion and the authors do not seem to have tried to experiment with different settings more before finalizing their approach. More concretely, the authors could have tried non-sharing of parameters between query and target graphs to see if it helps. Additionally, the authors could have experimented with alternatives for their Graph Isomorphism Network (GIN) module. Also why did the authors specifically choose sum pool rather than other pooling techniques. The authors could also have tried approximate nearest neighborhood (ANN) based approaches in their work.

[3] Khoa D Doan, Saurav Manchanda, Suchismit Mahapatra, and Chandan K Reddy. Interpretable graph similarity computation via differentiable optimal alignment of node embeddings. pages 665–674, 2021.
[4] Zhaoyu Lou, Jiaxuan You, Chengtao Wen, Arquimedes Canedo, Jure Leskovec, et al. Neural subgraph matching. arXiv preprint arXiv:2007.03092, 2020.
[5] Yujia Li, Chenjie Gu, Thomas Dullien, Oriol Vinyals, and Pushmeet Kohli. Graph matching networks for learning the similarity of graph structured objects. In International conference on machine learning, pages 3835–3845. PMLR, 2019
[6] Indradyumna Roy, Venkata Sai Velugoti, Soumen Chakrabarti, and Abir De. Interpretable neural subgraph matching for graph retrieval. AAAI, 2022.

---

> ### Author Response · Authors · 2022-08-02
> **Response to Reviewer ap6M**
>
> **q1. Why did you not cite and compare your approach against the following state-of-the-art recent approaches i.e., GOTSim [3], NeuroMatch [4], Graph embedding network (GEN) [5], and IsoNet [6] ?**
>
> *Response:* We would like to humbly state that this comment is partially incorrect.
>
> * We have cited GOTSim ([13]), NeuroMatch ([32]), and GEN ([27]).
> * We have empirically compared it with NeuroMatch and shown significantly superior performance (See Table 2b). As we have already illustrated in Baselines para of Sec 4.1, NeuroMatch is an algorithm for predicting subgraph isomorphism and not edit distance or subgraph edit distance. Hence, it can only be used for k-NN queries. The issue is the same for IsoNet, hence it was not compared against. We have now cited IsoNet in the revised version (highlighted in blue font in Sec 1).
>
> Additional baselines:
>
> * We do not compare with GEN (referred to as GMN in our paper), since SimGNN, H2MN and GENN-A* have compared with GEN and have shown better performance on the same datasets that we use. We compare with SimGNN, H2MN and GENN-A* and show better performance. Hence, comparing with GEN is redundant.
>
> * Regarding GotSIM, we did not compare since GotSIM does not generalize to SED; there is an explicit assumption of modeling a symmetric distance function because it uses cosine similarity to compare node neighborhoods. In addition, the denominator in Eq. 6 in GotSIM is also based on the assumption of whole-graph matching. In GED, H2MN is more recent and reported lower errors on average than GotSIM. Nonetheless, we have now compared with GotSIM in GED and show that GREED is indeed better. For GotSIM, we use the code provided by the authors. The results are below. This comparison has also been added to Table 1a in the revised version.
>
>
> RMSE of Greed vs GotSIM:
> | Method |Aids'| Linux |  IMDB |
> |--| -------- | -------- | -------- |
> |Greed |0.796 | 0.416| 6.734    |
> |GotSIM |  0.996   | 0.574     |  37.831    |
>
>
> **q2. Did you try any other alternatives to siamese graph neural network settings i.e., non-sharing of parameters between query and target graphs to see if it helps?**
>
> *Response:* This empirical analysis was already included in our submission. Specifically, in Sec 4.5 we compare GREED with GREED-dual where the weights between GNNs embedding the target and query graphs are not shared. As visible in Fig. 4i-4k, GREED is consistently better than GREED-dual. Furthermore, in Sec. 3.2, we also discuss the advantages that a Siamese architecture provides.
>
> **q3. Did you try any alternatives to the Graph Isomorphism Network (GIN) module?**
>
> *Response:* This experiment was also already included in the paper. Specifically, we refer to this ablation study in that last line of Section 4.5 with details in App G.
>
> **q4. Why did you use sum pool specifically?**
>
> *Response:*  An ablation study with sum-pool was already included in our submission. Specifically, we refer to this ablation study in the last line of Sec 4.5 with details in App G.
>
> Sum-pool can better distinguish graph sizes better than other aggregation functions such as mean-pool or max-pool. To elaborate, let us consider a graph $G_1$ that is significantly larger than another graph $G_2$. In this scenario, the individual coordinates of $G_1$’s embedding can potentially be significantly larger than those of $G_2$ since in $G_1$ the summation is being done over a larger set of embeddings. Both mean-pool and max-pool fail to capture the size information as effectively, since the max and the mean operations do not scale with the number of inputs.
>
> In the literature on graph similarity learning, SimGNN [2], GMN [24], and GENN-A\* [31] use attention-weighted sum-pool. We have not used attention as we observed in our development phase experiments that sum-pool outperforms attention-pool in our setting. Instead, we have utilized multi-granular summarization by utilizing layer-wise concatenation to obtain a rich representation of the graph structure. This approach has been shown to be effective in extracting useful subgraph features (see Xu et. al., “Representation Learning on Graphs with Jumping Knowledge Networks”, ICML 2018)
>
> **Appeal to the reviewer:** As we have clarified in our rebuttal, most of the questions asked were already present in the paper. In addition, we have now also included GotSIM. If the reviewer feels satisfied with our work and effort, we humbly appeal to the reviewer to raise our rating.

---

> > ### Author Response · Authors · 2022-08-06
> > **Looking forward to your post-rebuttal feedback**
> >
> > Thanks again for your insightful suggestions and comments. As the deadline for discussion is approaching, we are glad to provide any additional clarifications that you may need.
> >
> > In our previous response, we have carefully studied your comments and addressed the concerns raised. We summarize our responses with regards to the following aspects:
> >
> > 1. We clarified that all citations with the exception of IsoNet were already cited. We also pointed out that NeuroMatch, which does subgraph isomorphism tests, was already one of the baselines in k-NN queries (this is only setting where it can be used since it does not output a distance or perform whole-graph matching. NeuroMatch performs approximate subgaph-isomorphism tests, whereas our focus is on learning distances.) IsoNet suffers from the same limitation as NeuroMatch. We have now cited IsoNet.
> >
> > 2. We included GotSIM for GED and showed Greed is more accurate. We also explained why GotSIM does not extend to SED.
> >
> > 3. We explained that several of the requested experiments such as (i) Why siamese and its benefits, (ii) Why GIN and (iii) why sum-pool were already included in the submission.
> >
> > We hope that the additional explanations and comparison with GotSIM have convinced you of the merits of our work. On the whole, we show Greed is more accurate than a host of baselines and orders of magnitudes faster. Our codebase is also public for anyone to reproduce the results.
> >
> > Please do not hesitate to contact us if there are other clarifications or experiments we can offer.
> >
> > Thank you for your time again!
> >
> > Best,
> >
> > Authors

---

> > > ### Author Response · Authors · 2022-08-08
> > > **Looking forward to feedback by Reviewer ap6M**
> > >
> > > Dear Reviewer ap6M,
> > >
> > > We thank you for taking the time to provide critical comments on our work. We had added the comparison to GotSIM, cited IsoNet and clarified that all other issues were already addressed in our initial submission.
> > >
> > > With these additional experiments and clarification, we believe the Reviewer would now find the manuscript acceptable. Since today is the last day for author-reviewer discussions, please let us know if there are other comments for which we can provide any clarifications.
> > >
> > > We are looking forward to your feedback.
> > >
> > > Thank you,
> > >
> > > Authors

---

> > > > ### Comment · Reviewer_ap6M · 2022-08-08
> > > > **Thank you for your feedback**
> > > >
> > > > Hello dear authors,
> > > > Thank you for your comments and addressing the issues pointed out by all reviewers. I am updating my score to weak accept.

---

### Official Review · Reviewer_r5mu · 2022-07-14

**Rating:** 6
**Confidence:** 4
**Soundness:** 3 good
**Presentation:** 3 good
**Contribution:** 3 good

**Summary:**

Graph and subgraph edit distance (GED and SED) are important mechanisms to measure the distance between graph pairs. However, it is NP-hard and thus hard for scalability. This paper proposes a framework based on the graph neural networks, named GREED, to learn the graph embedding for the edit distances.

Unlike the previous methods, GREED is able to preserve the essential theoretical properties (e.g., triangle inequality, non-negativity, etc.) of the MED as the metric distance function, and also can easily be adapted to SED. Therefore, the embeddings learned from the GREED are indexable, significantly improving query time.

Comprehensive experiments are conducted to validate the proposed method. GREED outperforms state-of-art methods across ten real-world datasets up to millions of edges and is faster than other methods.


**Questions:**

The following questions are mainly related to section 4.5:
1. It seems that all the ablation studies are related to the SED based on the caption of Fig. 3, do you have any experiments with the GED? Those experiments can also help to test the consistency of the results.
2. From Fig. 3(i)-(k), we can observe that the blue line (Greed-NN) has a better performance on PubMed and CiteSeer datasets. What is the possible reason for it?
3. Does the Pre MLP matter since you use the one-hot encoding? Meanwhile, a bit larger problem is whether all the component in Figure 2(b) is necessary. In other words, can we directly use GIN (or other GCNs) to replace Figure 2(b)?




**Limitations:**

In the paper, the authors have explicitly or implicitly pointed out a couple of limitations: (1) not considering the edge labels; (2) not having well generalizability.
For limitation (1), I encourage the authors to explore and take advantage of some edge-related GCN models. For limitation (2), I suggest the author do more analysis of the dataset properties to find the reason for the poor generalizability. For example, is it caused by the structure domain shift (e.g., small world, scale-free property, and so on) or feature difference?

To best of my knowledge, there is no potential negative societal impact.


**Strengths And Weaknesses:**

Strengths:
First of all, the paper illustrates the motivation, problem, and previous works' limitations clearly. The paper's overall organization is good and easy for readers to follow. The authors open the source code, which is good for reproduction and will benefit the entire community.
Secondly, the proposed method (GREED) can learn the theoretical properties-preserved graph embedding. And the author provides the math proof for it.
Thirdly, the learned embedding is indexable and pair-independent, which is essential for scalability and fails most of the previous works in the literature. In this aspect, the author not only provides the neural networks-based model but also details of query algorithms.
Last but not least, lots of comprehensive experiments and discussions are conducted to validate the GREED, including prediction accuracy, method efficiency, scalability, and generalizability.

Weaknesses:
1. The main concern I have is the novelty of the design of the neural network and the corresponding ablation study (section 4.5). It seems that the second and third strengths above (i.e., preservation of GED's theoretical properties) are mainly from the last layer (i.e., GED and SED prediction functions F). In other words, the paper doesn't have theoretical insights into the choice of siamese architecture and the GNN component. In this case, the ablation study is important to support the claim. However, from Fig. 3(i)~3(k), the model component design seems not convincing (i.e., the green line is not the best). To better understand it, I raised a couple of questions in the "Questions" box.
2. Meanwhile, some writing parts can also be improved. For example,
(1) The table's index is unclear. There are two different "table 1" on page 7 and 15, respectively.
(2) Figure 4d should be "Dblp" instead of "IMDB".
(3) Try to put the figure (tables) and corresponding text on the same page for reading-friendly purposes.

---

> ### Author Response · Authors · 2022-08-02
> **Response to Reviewer r5mu**
>
> **It seems that all the ablation studies are related to the SED based on the caption of Fig. 3, do you have any experiments with the GED? Those experiments can also help to test the consistency of the results.**
>
> *Response:* We have added the same experiment on GED now (Fig. F in Appendix, which is referred to from Sec 4.5 in the main paper). The obtained data is produced below in the form of a table. As visible, the trends are similar. As more training data is provided, the gap between Greed and Greed-NN reduces. Greed-dual, i.e., the non-siamese version is consistently inferior to Greed.
>
>
> **RMSE in AIDS'**
> #Training Pairs|100000 | 10000|   1000|
> |-------------|--------|------|-------|
> |Greed| 0.8    | 0.86    |3.11|
> |Greed-NN| 0.95   | 1.23    |3.22|
> |Greed-Dual|1.23   | 3.47    |4.38|
>
> **RMSE in Linux:**
> #Training Pairs|100000 | 10000|   1000|
> |-------------|--------|------|-------|
> |Greed|0.42   | 0.40    |1.75|
> |Greed-NN|0.43    |0.55    |1.85|
> |Greed-Dual|0.52    |1.06    |2.21|
>
>
> **RMSE in IMDB:**
> #Training Pairs|100000 | 10000|   1000|
> |-------------|--------|------|-------|
> |Greed|6.73   | 4.77    |54.24|
> |Greed-NN|7.8    | 15.29   |647.97|
> |Greed-Dual|9.39   | 187.62  |285.36|
>
> **From Fig. 4(i)-(k), we can observe that the blue line (Greed-NN) has a better performance on PubMed and CiteSeer datasets. What is the possible reason for it?**
>
> *Response:* Greed-NN sometimes outperforms Greed *only* when the training data is very large. As visible in 4i-k, the blue line is below green only on the right part (larger training data) of the plot. We produce below the text from Sec 4.5 verbatim that explains why we observe this behavior.
>
> >  Compared to Greed, Greed-NN achieves marginally better performance at larger train sizes in Pubmed and Citeseer. However, in DBLP, Greed is consistently better. The number of subgraphs in a dataset grows exponentially with the node set size. Hence, an MLP needs growing training data to accurately model the intricacies of this search space. In DBLP, even 100k pairs is not enough to improve upon $\mathcal{F}$. Furthermore, since computing GED and SED is NP-hard, generating large volumes of training data is not desirable. Overall, these trends indicate that $\mathcal{F}$ enables better generalization and scalability with respect to accuracy. Furthermore, given that its performance is close to an MLP even on large training data, and it enables indexing, the benefits outweigh the marginal reduction in accuracy.

---

> > ### Author Response · Authors · 2022-08-02
> > **Part 2**
> >
> > **Does the Pre MLP matter since you use the one-hot encoding? Meanwhile, a bit larger problem is whether all the component in Figure 2(b) is necessary. In other words, can we directly use GIN (or other GCNs) to replace Figure 2(b)?**
> >
> > * *Pre-MLP:* The dimension of the one-hot vector increases linearly with the number of labels in a graph database and hence can be very large. The primary job of the pre-mlp is to reduce it to a desirable dimension size. Indeed, the same effect may also be obtained by directly feeding it to the first layer of GIN. However, since a GIN constructs embeddings by incorporating both structure and label information, one may desire a different dimensionality in the GIN layers. Hence, we have this separation. To summarize, the pre-mlp allows a more clear conceptual separation of the label-based embeddings from the subsequent structure+label embeddings. In terms of pure results, we do not observe a statistically significant difference in performance. We appreciate this observation and have added the above discussion to Sec 3.1. In addition, we have also conducted the ablation study to observe the RMSE of Greed with and without the pre-mlp layer. The results have been added to Appendix G. The results table is also re-produced below.
> >
> > |RMSE	|With Pre-MLP	|Without Pre-MLP|
> > |-------|---------------|----------------|
> > |AIDS (SED)	|0.51|	0.51|
> > |Amazon|	0.5o	|**0.39**|
> > |CiteSeer	|0.52|	**0.51**|
> > |Cora_ML	|**0.64**|	0.68|
> > |AIDS' (GED)|	**0.8**	|0.85|
> > |IMDB (GED)	|**6.73**	|7.68|
> > |LINUX (GED)|	0.42|	**0.41**|
> > |Protein	|0.52|	0.52|
> > |PubMed|	0.73|	0.73|
> >
> > * *Directly using GIN:* A GIN produces embeddings for each node. In our problem, we need an embedding for a graph. Hence, we need some aggregation function that can take a set of node embeddings and generate a single embedding characterizing the graphs. For that we use a sum-pool followed by an MLP.
> > * *Is the GIN important?:* Ablation study on replacing the GIN with other GNNs like a GCN or GAT is provided in Appendix G and Table B. GIN provides the best performance. This result is not surprising since GIN is provably more expressive than GCN or GAT in distinguishing graph structures (essential to SED or GED computation) and is as powerful as the Weisfeiler-Lehman Graph Isomorphism test.
> >
> >
> > **It seems that the second and third strengths above (i.e., preservation of GED's theoretical properties) are mainly from the last layer (i.e., GED and SED prediction functions F). In other words, the paper doesn't have theoretical insights into the choice of siamese architecture and the GNN component. In this case, the ablation study is important to support the claim.**
> >
> > *Response:* The siamese architecture is also necessary to ensure metric and triangle inequality of GED and SED respectively. The first para of Sec 3.2 discusses this aspect. We reproduce it verbatim below.
> >
> > > If we do not use a siamese architecture, then the embedding model for the query and the target graphs would be different. Hence, the predicted distance would violate symmetry, and therefore, would not be a metric. Furthermore, if the distance computations are pair-dependent [27, 42, 2], i.e., it jointly learns the embedding of the query and the target, then a single graph may correspond to multiple representations. Hence, it would not be a metric or satisfy the triangle inequality.
> >
> > In addition, we also refer to Sec 4.5, where we compare GREED with its non-siamese version, GREED-dual, where the weights are not shared. As visible in Fig. 4i-4k, GREED is consistently better than GREED-dual.
> >
> >
> > **Meanwhile, some writing parts can also be improved. For example, (1) The table's index is unclear. There are two different "table 1" on page 7 and 15, respectively. (2) Figure 4d should be "Dblp" instead of "IMDB". (3) Try to put the figure (tables) and corresponding text on the same page for reading-friendly purposes.**
> >
> > *Response:* We apologize for these presentation issues. They occurred due to latex reference errors and skipped our notice. All of these issues have now been corrected.

---

> > > ### Author Response · Authors · 2022-08-06
> > > **Looking forward to your post-rebuttal feedback**
> > >
> > > Thanks again for your insightful suggestions and comments. As the deadline for discussion is approaching, we are glad to provide any additional clarifications that you may need.
> > >
> > > In our previous response, we have carefully studied your comments and addressed the concerns raised. We summarize our responses with regards to the following aspects:
> > >
> > > 1. We conducted additional ablation studies to show that (i) the trends with respect to the non-siamese (Greed-dual) and MLP (Greed-NN) versions also hold for GED, (ii) clarify our position on the pre-mlp module including new empirical data.
> > >
> > > 2. We have addressed all the issues related to presentation. The changes are highlighted in blue in the revised version.
> > >
> > > 3. We explained why the siamese architecture is required for the theoretical guarantees we provide with respect to preservation of distance properties and its implication on the accuracy with actual empirical data (now further boosted with experiments on GED as well).
> > >
> > > We hope that the provided new experiments and additional explanations have convinced you of the merits of our work. Please do not hesitate to contact us if there are other clarifications or experiments we can offer.
> > >
> > > On the whole, we show Greed is more accurate than a host of baselines and orders of magnitudes faster. Our codebase is also public for anyone to reproduce the results.
> > >
> > > Thank you for your time again!
> > >
> > > Best,
> > >
> > > Authors

---

### Author Response · Authors · 2022-08-02
**General comments on the rebuttal. Applicable to all reviewers**

We thank the reviewers for the critical comments and suggestions. Please find a point-by-point response to the comments raised by the reviewers below. We have also updated the main manuscript and the appendix to address these comments. The changes made in the manuscript are highlighted in **blue color**. *All references to figures, tables, sections, citations, etc. made in our response are based on the updated version.*

---

### Meta-Review · Area_Chair_E6EK · 2022-08-30

**Recommendation:** Accept
**Confidence:** Certain

**Metareview:**

The paper studies an important problem of computing distance functions across graphs which is NP-hard to solve in the worst case. The author provide a theoretical analysis of certain properties of the algorithm, and show its relevance in practice. The reviewers pointed out some weakness, but the rebuttal helped resolve some of those.  Please address those comments for the camera ready version.

The paper has weak accept votes, but in light of the significance of the topic, I also lean toward accepting the paper.

**Award:**

No

---

### Decision · Program_Chairs · 2022-09-14

Accept